# SHARPNESS-AWARE GEOMETRIC DEFENSE FOR ROBUST OUT-OF-DISTRIBUTION DETECTION

## ABSTRACT

Out-of-distribution (OOD) detection ensures safe and reliable model deployment. Contemporary OOD algorithms using geometry projection can detect OOD or adversarial samples from clean in-distribution (ID) samples. However, this setting regards adversarial ID samples as OOD, leading to incorrect OOD predictions. Existing efforts on OOD detection with ID and OOD data under attacks are minimal. In this paper, we develop a robust OOD detection method that distinguishes adversarial ID samples from OOD ones. The sharp loss landscape created by adversarial training hinders model convergence, impacting the latent embedding quality for OOD score calculation. Therefore, we introduce a **Sharpness-aware Geometric Defense (SaGD)** framework to smooth out the rugged adversarial loss landscape in the projected latent geometry. Enhanced geometric embedding convergence enables accurate ID data characterization, benefiting OOD detection against adversarial attacks. We use Jitter-based perturbation in adversarial training to extend the defense ability against unseen attacks. Our SaGD framework significantly improves FPR and AUC over the state-of-the-art defense approaches in differentiating CIFAR-100 from six other OOD datasets under various attacks. We further examine the effects of perturbations at various adversarial training levels, revealing the relationship between the sharp loss landscape and adversarial OOD detection. The implementation code will be released upon paper acceptance.

## 1 INTRODUCTION

Advancements in artificial intelligence (AI) go beyond mere model accuracy. One critical aspect is the AI model's capability to identify and reject unfamiliar samples, ensuring reliable AI deployment. The technical field of detecting *out-of-distribution (OOD)* samples [45, 53] has raised substantial attention. The aim is to distinguish disjoint OOD samples from the in-distribution (ID) training samples. For example, an image classifier should recognize unfamiliar input images outside training classes to avoid generating unreliable predictions.

The deep neural network is known to be vulnerable to *adversarial attacks* [18], which are intentionally manipulated perturbations in a subtle way that is malicious to mislead model predictions. A handful of adversarial defense studies are proposed to secure the model prediction against the attacks [40, 38, 15]. Notably, *adversarial training* and *hyperspherical geometry learning* effectively alleviate adversarial situations in image classification tasks. In the case of OOD detection, existing studies typically predict adversarial samples as OOD samples [29], leading to substantial alarms for adversarial ID cases shown in Figure 1. These studies have still not yet explored how to distinguish adversarial ID from adversarial OOD samples and thus are still not resilient to attacks in realizing real OOD applications. We aim to ensure the OOD detection system robustly operates in clean and adversarial conditions.

The task of differentiating OOD itself is hard due to the widespread new data pattern to the model [16], and suffering from adversarial attacks increases the complexity of OOD detection. ATOM is a pioneering framework for dealing with attacks on open-set samples [5]. Recently, Azizmalayeri *et al.* [1] found that adversarial attacks on both ID and OOD data significantly degrade detection accuracy. They introduced an Adversarial Training Discriminator (ATD) with an outlier exposure strategy that simulates both adversarial ID and OOD samples. The outlier exposure method highly depends on the auxiliary OOD datasets which are expected to be excessively large. This requirement

leads to inefficient and impracticality in real-world applications. We target the defense against challenging *white-box* attacks on both ID and OOD data and seek effective perturbation strategies without relying on additional large outlier datasets.

In this work, we aim to tackle this robust OOD detection issue by combining two perspectives, including *geometry optimization* and *loss landscape smoothing*. First, the hypersphere and hyperbolic geometries can learn compact representations for OOD detection [36], but we still empirically observe a high false positive rate. Therefore, we further examine the ability of the multiple-geometric learning method [30] in accommodating ID data variability under adversarial attacks. Second, sharp loss has been observed in prior research [14, 56] which is caused by adversarial samples. These samples increase the gradient norm and the

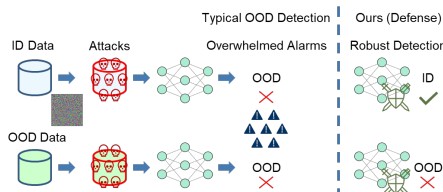

Figure 1: Problem scope of the adversarial OOD detection

subsequent local minimum, sharp loss, and challenges in convergence. The GAN-like structure in ATD is also known for its loss convergence issues.

Therefore, we introduce a sharpness-aware adversarial training framework that effectively alleviates the sharp loss landscape, achieving robust latent geometry learning. Our backbone network learns a Multi-Geometry Projection (MGP) [30] by incorporating two Riemannian (hypersphere and hyperbolic) geometries with distinct curvatures to fully characterize the complex ID data. In the adversarial training procedure, we propose to utilize the Riemannian Sharpness-aware Minimization (RSAM) [55, 47], which improves the multiple Riemannian geometry convergence by flattening the adversarial loss landscape. We empirically find that performing adversarial training based on the *Jitter* attacks [41] demonstrates generalizability in defending against other attacks.

Our experiments comprehensively investigate mainstream OOD detection approaches with and without adversarial training. We use CIFAR-10 and CIFAR-100 as ID datasets and perform OOD evaluation using six other datasets. We compare the proposed SaGD against ATD [1], the state-of-the-art (SoTA) defense approach for OOD detection, and show our improvements. Additionally, we examine the effects of different adversarial training approaches to reveal the generalization ability of SaGD in defending other types of attacks. In contrast to other OOD studies [1, 5, 6], which only present one type of Projected Gradient Descent (PGD) attack, our results are comprehensively from the average of six conditions, including the case without attack and five other cases under different attacks. We report the area under the ROC curve (AUC) along with the false positive rate at 95% true positive rate (abbreviated as $FPR_{95}$) as evaluation metrics. The $FPR_{95}$ is a common metric for OOD detection; however, it is not reported in [1]. In the adversarial OOD detection experiments using the CIFAR-10 ID dataset, our SaGD robustly reduces 14.91% $FPR_{95}$ and enhances 7.47% AUC over the SoTA approach. We also achieved a 17.71% average $FPR_{95}$ reduction and 10.18% AUC improvement using CIFAR-100 as the ID dataset.

Our contribution is summarized as follows:

- We introduce a novel sharpness-aware method for improving OOD detection in adversarial training. Our method investigates the combination of Riemannian geometries under adversarial conditions. This expansion of geometry space sharpens our defense against adversarial attacks and avoids reliance on large OOD datasets for auxiliary training.

- We examine different perturbation techniques (not limited to PGD) for adversarial training to identify their effectiveness for robust OOD detection.

- We investigate various adversarial attacks on different OOD detection approaches and report results on $FPR_{95}$ and AUC. Our SaGD sets a new SoTA for OOD detection, excelling in $FPR_{95}$ and AUC metrics, both with or without attacks.

- We analyze the relations between the minimization of a sharp loss landscape and OOD detection performance under various adversarial conditions.

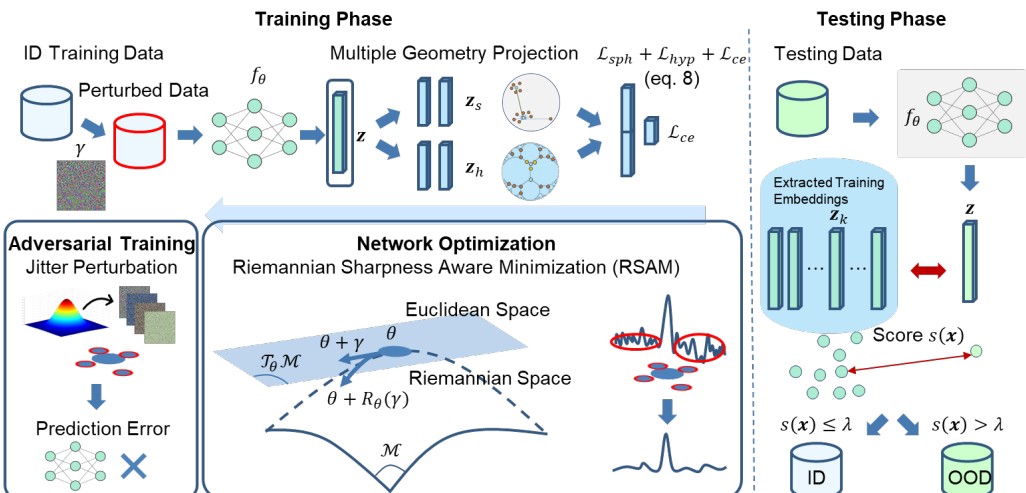

Figure 2: Overview of the proposed *Sharpness-aware Geometry Defense (SaGD)* framework for robust OOD detection. The Multi-Geometry Projection (MGP) network is trained using Jitter-based adversarial samples and optimized via sharpness-aware loss minimization using RSAM. In testing, sample embedding is computed for scoring to discern OOD from ID cases.

## 2 RELATED WORK

### 2.1 OOD DETECTION

**Post-processing OOD detection.** Model-agnostic OOD detection methods [45, 53] formulate scoring functions based on prediction probability and energy score. Determining prediction confidence can take various forms, such as utilizing softmax outputs [21], energy-based scores [33], or entropy functions [4]. To avoid re-training or excessive tuning of the given model, recent advancements focus on introducing perturbation [31], conducting pruning [12], and generating an unknown novel class [48] to enhance the distinction between OOD and in-distribution (ID) samples.

**Model training OOD detection.** Other OOD studies have sought to enhance fixed-model post-processing by incorporating network constraints during training to improve OOD detection. Sophisticated designs are devised for network space projection and embedding distance measurement to effectively train models for OOD detection. Noteworthy examples include SSD [42] and KNN+[46], employing contrastive loss for latent embedding learning and calculating Mahalanobis[29] and non-parametric KNN distances, respectively. A recent addition to this line of work is the CIDER framework [36], which has demonstrated improved OOD detection performance by imposing constraints on samples using a hypersphere-based loss function. The hyperbolic embedding also demonstrates the enhanced ability for OOD detection [19]. Despite the impressive results achieved by these OOD detection approaches, their performance is not robust when facing adversarial samples in practice.

### 2.2 ADVERSARIAL DEFENSES

Adversarial training [34] stands out as a key defense against adversarial attacks. This method involves integrating adversarial samples into model training to bolster the network's resilience against perturbations. The goal is to approximate potential perturbations in adversarial samples, using them to enhance model accuracy. However, the incorporation of adversarial samples unavoidably leads to a degradation in prediction accuracy due to the introduction of various noises. To this end, Helper-based Adversarial Training (HAT) seeks balance and reduces harm from adversarial samples by tailoring network architecture and loss designs [40]. Notably, works such as [43, 44] train dual-attentive denoising layers, leading to clean reconstructed samples from adversarial ones. Originally devised for addressing the open-set detection problem, these techniques find application in OOD detection scenarios under adversarial attacks [1].

Outlier exposure [22, 39] emerges as a strategy in OOD detection, broadening its capabilities by incorporating outliers during training. Although these techniques can also foster a robust learning space for adversarial outliers [6], their practical utility is constrained by the uncertainty surrounding the optimal inclusion of outliers and the types of adversarial that should be artificially introduced.

## 2.3 REGULARIZATION AND ADVERSARIAL ROBUSTNESS

Regularizing neural networks proves effective against adversarial attacks by preventing the adoption of overly complex parameters, and avoiding suboptimal convergence at saddle points. When attacked samples have distributions vastly different from the training space, it significantly biases the network parameters. Implementing regularization based on various network designs, such as angular and margin regularization on hypersphere geometry, enhances adversarial robustness [38, 15]. Prioritizing more regularization for vulnerable samples minimizes the robustness risk and improves generalizability [52]. The underlying reason is that adversarial training can generate rugged sample space and thus hinder model convergence. Such a sharp loss landscape hinders the training process with scattered gradients and increased curvatures [32].

Sharpness-Aware Minimization (SAM) [17] is a well-known technique for its regularization ability to mitigate training overfitting on a sharp loss landscape. A recent study [49] delves into SAM's potential for adversarial robustness and empirically establishes a lightweight alternative to PGD adversarial training without significant sacrifices in clean sample accuracy. However, the integration of SAM-based regularization with adversarial training, especially in OOD detection, remains limited. The exploration of geometric projection associations, such as RSAM operating on the Riemannian manifold [55, 47], is largely uncharted. This paper advances current research by combining RSAM with multi-geometry learning techniques for OOD detection. We also investigate the effects of various adversarial training types in experiments.

## 3 SHARPNESS-AWARE GEOMETRY DEFENSE (SAGD)

Figure 2 overviews our proposed SaGD framework, where the training phase consists of adversarial training using Jitter-based adversarial samples (§3.1), multi-geometry projection (§3.2), and sharpness-aware optimization (§3.3). The multi-geometry backbone combines the hypersphere and hyperbolic branches in a multi-task joint loss optimization scheme. We first introduce the architecture along with the scoring function for OOD score calculation. We then show how the Riemannian Sharpness-aware Minimization (RSAM) optimizes the framework with adversarial training.

**Problem setup.** Given labeled data $(x, y)$ from a distribution $\mathcal{D}$, we consider a model $f_\theta$ with parameters $\theta$. The training and testing data are denoted as $\mathcal{D}_s$ and $\mathcal{D}_t$, respectively, where $\mathcal{D}_t$ contains both in-distribution ($\mathcal{D}_{id}$) and out-of-distribution ($\mathcal{D}_{ood}$) test data. We assume $\mathcal{D}_{id}$ is drawn from the same distribution as $\mathcal{D}_s$, while $\mathcal{D}_{ood}$ is from a different distribution that needs to be distinguished. The standard procedure for OOD detection is as follows: (1) Train a model $f_\theta$ with $\mathcal{D}_s$. (2) Fix model parameter $\theta$ during test time. For each test sample $x$, derive embedding $z$ using $f_\theta$. (3) Calculate OOD score $s(x)$ and differentiate OOD samples with a threshold $\lambda$. To protect the model against adversarial attacks, we focus on the first step to strengthen the model's robustness.

### 3.1 ADVERSARIAL TRAINING

We utilize the *Jitter* adversarial attack [41] to generate adversarial samples. Each input sample $x$ is perturbed by *Jitter* attack to simulate the inference-time attacks. Denote the perturbed samples as $x_\gamma = x + \gamma$, where $||\gamma||_p \leq \epsilon$ with an $l_p$-norm bound and we select $p$ to be the infinite norm.

The *Jitter* attack rescales the softmax function as $\hat{\mathbf{h}} = softmax\left(\alpha \cdot \frac{\mathbf{h}}{||\mathbf{h}||_\infty}\right)$. This is based on a finding that a small value range of output logits $\mathbf{h}$ can reduce the attack success rate. By default, $\alpha$ is chosen to be 10. Then, our optimization goal for the attacking model in adversarial training is to maximize the Euclidean distance between the rescaled softmax output $\hat{\mathbf{h}}$ and the one-hot encoded ground truth vector $\mathbf{y}$: $L_2 = ||\hat{\mathbf{h}} - \mathbf{y}||_2$. We further perturb the target loss by adding a Gaussian noise $\mathcal{N}(0, \sigma)$ with magnitude $\sigma$. Such perturbed attack loss is then: $L_{\mathcal{N}} = ||\hat{\mathbf{h}} + \mathcal{N}(0, \sigma) - \mathbf{y}||_2$.

An adaptive searching rule is designed to downscale the perturbation by a factor $\beta$ once the attack succeeds which avoids over-optimized adversarial samples biased far from ID characteristics. The *Jitter* loss is then:

$$L_{Jitter} = \begin{cases} \frac{||\hat{\mathbf{h}} + \mathcal{N}(0,\sigma) - \mathbf{y}||_2}{\beta} & \text{if } f_\theta(x_\gamma) = y, \\ ||\hat{\mathbf{h}} + \mathcal{N}(0,\sigma) - \mathbf{y}||_2 & \text{otherwise.} \end{cases} \quad (1)$$

## 3.2 Multi-Geometry Projection (MGP)

Our backbone network $f_\theta$ incorporates a dual-stream geometry projection to capture diverse latent structures in ID data. Each geometry stream is defined by its specific loss function for joint optimization. In this context, we introduce hypersphere and hyperbolic geometry, both Riemannian manifolds with positive and negative curvature, respectively. The curvature serves as an indicator of deviation from the Euclidean space. Hyperspherical geometry has shown its effectiveness in OOD detection [36]. The hyperbolic space has been used in open-set recognition [11] that can model the hierarchical structures found in real-world vision data [13], as evident in datasets like Imagenet.

We assume parameter $\theta$ resides on a Riemannian manifold $\mathcal{M}$ with the Riemannian metric tensor $g^\mathcal{M}$. The tensor $g^\mathcal{M} : \mathcal{T}_\theta \mathcal{M} \times \mathcal{T}_\theta \mathcal{M}$ consists of inner products in its tangent space $\mathcal{T}_\theta \mathcal{M}$. A retraction map $R_\theta$ provides transformations from $\mathcal{M}$ to the tangent space $\mathcal{T}_\theta \mathcal{M}$. The tangent space can be regarded as a measure of small deviation $\gamma$ near parameter $\theta$, and the metric $g^\mathcal{M}$ smoothly varies across $\theta \in \mathcal{M}$, resulting in the geodesic distance. The deviation $\gamma$ on $\mathcal{T}_\theta \mathcal{M}$ is considered as the perturbation generated for adversarial training (as discussed in §3.1), which will be utilized in Riemannian manifold optimization (§3.3).

We incorporate the following geometries, each with its own loss metric designs.

**Hypersphere geometry**: Learning hypersphere geometry involves compactness and disparity loss functions to group data samples onto a hypersphere. These functions ensure that samples from different classes are kept at sufficient distances from each other. The hypersphere projection approach initially introduced as CIDER [36], is based on the von Mises-Fisher (vMF) distribution assumption. It is calculated using a unit vector $\mathbf{z_s} \in \mathcal{R}_s^d$ in class $k$ and the class prototype $\boldsymbol{\mu}_k$ as: $p_d(\mathbf{z_s}; \boldsymbol{\mu}_k) = \tau \exp(\boldsymbol{\mu}_k \mathbf{z_s}/\tau)$, where $\tau$ is a temperature parameter. The probability of the embedding $\mathbf{z_s}$ assigned to class $k$ is: $\mathcal{P}(y = k|\mathbf{z_s}; \{\boldsymbol{\mu}_k, \tau\}) = \frac{\exp(\boldsymbol{\mu}_k \mathbf{z_s}/\tau)}{\sum_{j=1}^K \exp(\boldsymbol{\mu}_j \mathbf{z_s}/\tau)}$. We derive the *compactness loss* by taking negative log-likelihood, which compels the projected samples to stay near the class prototypes.

$$\mathcal{L}_{com} = -\frac{1}{N} \log \frac{\exp(\boldsymbol{\mu}_k \mathbf{z_s}/\tau)}{\sum_{j=1}^K \exp(\boldsymbol{\mu}_j \mathbf{z_s}/\tau)}. \tag{2}$$

The *disparity loss* penalizes the class prototypes that are too close to each other:

$$\mathcal{L}_{dis} = -\frac{1}{K} \sum_{i=1}^K \log \frac{1}{K-1} \sum_{j=1}^K \mathbf{1}_{ji} \exp(\boldsymbol{\mu}_i \boldsymbol{\mu}_j/\tau), \tag{3}$$

where $\mathbf{1}_{ji}$ is indication function, $\mathbf{1}_{ji} = \begin{cases} 1 & \text{if } j \neq i, \\ 0 & \text{otherwise.} \end{cases}$ The *hypersphere loss* function is $\mathcal{L}_{sph} = \mathcal{L}_{com} + \mathcal{L}_{dis}$, which imposes constraints on ID intra-class compactness and inter-class disparity on the hypersphere. Meanwhile, OOD data are more likely to be separated farther from ID prototypes.

**Hyperbolic geometry**: A hyperbolic space $H^d$ consists of $d$-dimensional Riemannian manifolds with constant negative curvature [25]. An isomorphic hyperbolic transformation, Poincaré Ball $(\mathcal{D}_c^d, g^\mathcal{D})$, defines a manifold $\mathcal{D}^d = \{\mathbf{u} \in \mathbb{R}^d : c||\mathbf{u}|| < 1\}$ equipped with the Riemannian metric $g^\mathcal{D}(\mathbf{u}) = (\lambda_\mathbf{u}^c)^2 g^E = (\frac{2}{1-c||\mathbf{u}||^2})^2 \mathbf{I}$, where $\lambda = \frac{2}{1-c||\mathbf{u}||^2}$ is a conformal factor with curvature $c$, and $g^E = \mathbf{I}$ is an Euclidean metric tensor. The manifold operates on Mobius gyrovector space with Mobius addition $\oplus_c$ and scalar multiplication $\otimes_c$ (referring to appendix A.1).

The pairwise geodesic distance is in the following form for two points $\mathbf{u}$ and $\mathbf{v}$: $D(\mathbf{u}, \mathbf{v}) = \frac{2}{\sqrt{c}} \text{arctanh}(\sqrt{c}|| - \mathbf{u} \oplus_c \mathbf{v}||)$. Utilizing the operations of the hyperbolic space, we project the latent embedding with a hyperbolic head to derive the embedding $\mathbf{u}$ on the Poincaré ball. Considering an augmented set $\mathcal{A}$ from $\mathcal{X}$ to form a full set $\mathcal{I} = \mathcal{A} \cup \mathcal{X}$, the supervised contrastive loss is calculated on the positive sample $p(i)$ of the $i \in \mathcal{I}$ in contrast to other augmented samples $a \in \mathcal{A}$. The supervised hyperbolic contrastive loss can thus be formulated as $\mathcal{L}_{hypb} =$

$$-\sum_{i \in \mathcal{I}} \frac{1}{|P(i)|} \sum_{p \in P(i)} \log \frac{\exp(-D(\mathbf{z}_i, \mathbf{z_{h}}_p)/\tau)}{\sum_{a \in \mathcal{A}} \exp(-D(\mathbf{z_{h}}_i, \mathbf{z_{h}}_a)/\tau)}.$$

The final loss is the combination of the hypersphere and hyperbolic losses, along with a cross-entropy loss $\mathcal{L}_{ce}$ to optimize for ID classification accuracy: $\mathcal{L} = \mathcal{L}_{sph} + \mathcal{L}_{hypb} + \mathcal{L}_{ce}$.

## 3.3 Riemannian Sharpness-aware Minimization

Learning complex latent geometries may face undesirable peaks in the loss minimization process. Inspired by SAM [17] which was originally crafted for model generalization, we employ an improved approach for Riemannian manifolds [55, 47] tailored to our multi-geometry network. The consideration of multiple geometries in the network represents various manifolds that might not consistently converge in the same gradient direction. The recent work [36] only accounts for a single hypersphere geometry, which limits the ability to represent the OOD space. In our scenario, we aim to utilize the Riemannian manifold optimization strategy to strenthen multiple geometries.

Given a loss function $\mathcal{L}(\theta)$ with model parameter $\theta \in \mathcal{M}$ and retraction map $R_\theta$, the *manifold sharpness* is defined as $\mathcal{L}_S = \max_{||\delta||^2_\theta \leq \rho} \mathcal{L}(R_\theta(\delta)) - \mathcal{L}(\theta)$, where $\delta$ is a projected perturbation in the tangent space $\mathcal{T}_\theta \mathcal{M}$ of the manifold $\mathcal{M}$. The minimization of $\min_{\theta \in \mathcal{M}} \mathcal{L}_S$ reduces loss sharpness.

We simplify the first term in $\mathcal{L}_S$ using Taylor expansion to approximate perturbed loss in the maximization process: $\mathcal{L}(R_\theta(\delta)) \approx \mathcal{L}(\theta) + \langle \nabla_\theta \mathcal{L}(\theta), \delta \rangle_\theta$, where $\nabla_\theta$ denotes the Riemannian gradient. A closed-form solution for $\mathcal{L}_S$ is picking $\delta$ equal to the Riemannian gradient within the upper bound $\rho$. The optimal perturbation is then $\delta^* = \rho \frac{\nabla_\theta(\mathcal{L}(\theta))}{||\nabla_\theta(\mathcal{L}(\theta)||_\theta}$. We project $\delta^*$ onto the tangent space via $R_\theta$ and derive the optimal parameter $\theta^* = R_\theta(\delta^*)$. The network parameter in the next training iteration $\theta'$ can be updated using Riemannian gradient descent as: $\theta' = R_\theta(-\eta \cdot \nabla_\theta(\mathcal{L}(\theta^*)))$, where $\eta$ is the learning rate. During the adversarial training described in §3.1, the sharpness $\mathcal{L}_S$ on the loss landscape would unexpectedly increase. Our solution is introducing RSAM, which can regularize the network to increase convergence quality.

## 3.4 OOD Scoring Function

With a trained network $f$ in the MGP framework, we extract the penultimate layer output as an L2 normalized embedding $\mathbf{z}$ of the sample $\mathbf{x}$ to compute its OOD score. To distinguish OOD from ID samples, we calculate the embedding distance between each input sample and the training ID samples and specify the $k^{th}$ nearest neighbor as a reference embedding $\mathbf{z}_k$. The OOD score is based on the L2 distance, $S(\mathbf{z}) = ||\mathbf{z} - \mathbf{z}_k||_2$. An OOD sample is detected using a threshold $\lambda$ on the score $S(\mathbf{z})$.

## 4 Experiments

**Dataset:** Our OOD detection experiments are categorized into results for approaches with and without defense. For OOD detection without adversarial defense, we use CIFAR-10 and CIFAR-100 [27] as the ID dataset, and evaluate the performance on six other datasets that are treated as OOD: Tiny-ImageNet [28], Place365 [57], LSUN [54], LSUN-Resize [54], iSUN [50], and Textures [8]. For the compared OOD detection with adversarial defense, ATOM [5] and ATD [1] requires Food-101 [2] dataset for additional outlier data training and SVHN [37] dataset for validation.

**Evaluation metric:** (1) FPR$_{95}$: False positive rate at true positive rate 95% in the Receiver Operating Characteristic (ROC) analysis. (2) AUC: Area under the ROC curve.

**Attack setup:** We investigate a set of attacks including PGD [34], FGSM [18], FAB [9], Jitter [41], and Carlini and Wagner Attack (CW) [3], which are implemented using the TorchAttacks toolbox [26]. The attacks are constrained with perturbation bound $\epsilon = \frac{8}{255}$ and step size $\frac{2}{255}$ for 10-step iterations.

**Model Configurations:** Our CIFAR-10 evaluation uses a ResNet-18 backbone network and CIFAR-100 uses ResNet-34. The base optimizer is stochastic gradient descent (SGD) with momentum 0.9, weight decay $10^{-4}$, and an initial learning rate of 0.5. This optimizer is regularized by RSAM in §3.3. The model undergoes training for 500 epochs with a batch size of 512. We specify the intermediate layer with 128 dimensions. The curvature $c$ of hyperbolic geometry is set to be 0.01.

### 4.1 Evaluation of out-of-distribution accuracy

We report the averaged OOD detection results over six OOD datasets. Our adversarial results are mean values of the averaged OOD results over five adversarial conditions. The full results for each dataset under different attacks are reported in the supplementary files.

Table 1: Evaluation of OOD detection methods **without adversarial training** and **with adversarial training**. We report the average $FPR_{95}$ and AUC scores across the six OOD datasets. Apart from the "Clean" setting, "Adversarial" conditions denote the further average results over five attacks (PGD, Jitter, FAB, FGSM, and CW). Complete results are presented in the supplementary files.

| | | | | | | | | |
|---|---|---|---|---|---|---|---|---|
| Without Adversarial Training | | | | | | | | |
| ID Dataset | CIFAR10 | | | | CIFAR100 | | | |
| Condition | Clean | | Adversarial | | Clean | | Adversarial | |
| Metric | $FPR_{95}\downarrow$ | $AUC\uparrow$ | $FPR_{95}\downarrow$ | $AUC\uparrow$ | $FPR_{95}\downarrow$ | $AUC\uparrow$ | $FPR_{95}\downarrow$ | $AUC\uparrow$ |
| KNN+ | 18.06 | 96.59 | 67.14 | 81.25 | 65.47 | 85.07 | 90.47 | 56.87 |
| SSD | 33.08 | 94.87 | 69.11 | 72.88 | 70.98 | 84.94 | 90.56 | 54.53 |
| CIDER-KNN | 52.20 | 88.41 | 65.18 | 78.39 | 65.99 | 83.44 | 72.88 | **82.01** |
| CIDER-Maha | 51.19 | 88.91 | 55.52 | 85.87 | 67.28 | 84.36 | **68.40** | 80.03 |
| MGP-KNN | **21.60** | **96.11** | 69.08 | 79.49 | **57.89** | **85.26** | 73.02 | 77.52 |
| MGP-Maha | 29.98 | 95.36 | **47.41** | **90.23** | 66.47 | 83.19 | 73.80 | 72.62 |
| With Adversarial Training | | | | | | | | |
| Condition | Clean | | Adversarial | | Clean | | Adversarial | |
| Metric | $FPR_{95}\downarrow$ | $AUC\uparrow$ | $FPR_{95}\downarrow$ | $AUC\uparrow$ | $FPR_{95}\downarrow$ | $AUC\uparrow$ | $FPR_{95}\downarrow$ | $AUC\uparrow$ |
| ACET | 33.80 | 83.96 | 69.90 | 64.48 | 80.04 | 75.43 | 83.19 | 75.43 |
| CCU | 26.46 | 86.03 | 64.66 | 67.99 | 79.63 | 77.46 | 81.67 | 77.26 |
| ATOM | 23.57 | 88.22 | 56.13 | 79.08 | 72.81 | 79.31 | 76.82 | 80.12 |
| ATD | 27.46 | 94.15 | 42.59 | 87.36 | 63.04 | 82.90 | 67.58 | 77.41 |
| SaGD | **22.46** | **95.77** | **27.68** | **94.83** | **50.89** | **87.26** | **49.87** | **87.59** |

**Without Defense**: The upper part of Table 1 showcases popular OOD detection methods under adversarial attacks. We compare with embedding-based methods without adversarial training including SSD [42], KNN+ [46], and CIDER [36], and our MGP approach (§3.2). We consider both KNN [46] and Mahalanobis [29] as scoring functions for CIDER and MGP, which are denoted in the form of 'detector-function' in Table 1. Other score-based methods along with detailed results are reported in supplementary files. These OOD approaches are not designed to defend against malicious attacks. Thus, the experiment can reflect performance degradation under attacks.

MGP-Maha outperforms other methods on the CIFAR-10 in Table 1. CIDER-Maha still obtains an 8.11% gap in $FPR_{95}$ though the 68.40% $FPR_{95}$ is notable on the CIFAR-100. In the context of CIDER and MGP detectors under adversarial attacks, Mahalanobis scores (Maha) [29] stands out as a remarkable scoring function. Conversely, KNN generally performs well in clean conditions.

**With Defense**: In Table 1, we compare our proposed SaGD approach to the SToA adversarial defense methods, ATD [1]. Other methods including ACET [20], CCU [35], and ATOM [5] proposed in similar but different settings are also reported. SaGD achieves notable performance with average $FPR_{95}$ of 27.68% and AUC of 94.83% using CIFAR-10 as ID data. For CIFAR-100 as the ID data, SaGD attains an average $FPR_{95}$ of 50.03% and an AUC of 87.53%, outperforming ATD significantly. The confidence-based algorithms such as CCU and ACET are not resilient to adversarial conditions, with average $FPR_{95}$ values over 60% and 80% for the CIFAR-10 and CIFAR-100 datasets. Although ATOM obtains a 23.57% average $FPR_{95}$ in clean OOD detection using the CIFAR-10 ID data, the adversarial results are still inferior to SaGD and ATD. SaGD demonstrates substantial superiority over ATD by at least 17% on the CIFAR-100 ID dataset. For the clean set without attacks, ATD achieves a relatively close AUC to SaGD on the CIFAR-10 dataset but falls short by 4.64%. Notably, the difference in $FPR_{95}$ is substantial, with SaGD achieving 5.00% and 12.15% lower $FPR_{95}$ than ATD on CIFAR-10 and CIFAR-100 datasets, respectively. The more difficult OOD detection conditions of CIFAR-100 reveal even more pronounced advantages of using SaGD.

Another advantage of our SaGD is its ability to circumvent the need for additional outlier datasets, a requirement in ATD and ATOM for performing outlier exposure.

## 4.2 ABLATION STUDY

We conduct an ablation study for related techniques using the CIFAR-10 ID dataset, to elucidate the effects of each module in our SaGD framework. The results are presented in Table 2.

Table 2: Ablation study results on CIFAR-10. The upper part presents the ablation of modules in SaGD including MGP/CIDER network, Jitter adversarial training, and RSAM optimization. Our proposed SaGD is located in the last row of the upper table (MGP+RSAM+Jitter). The lower part is about replacing Jitter with other perturbations for adversarial training.

| CIDER | | Clean | | PGD | | Jitter | | FAB | | FGSM | | CW | | Average | |
|---|---|---|---|---|---|---|---|---|---|---|---|---|---|---|---|
| RSAM | AT | $FPR_{95}$ | AUC | $FPR_{95}$ | AUC | $FPR_{95}$ | AUC | $FPR_{95}$ | AUC | $FPR_{95}$ | AUC | $FPR_{95}$ | AUC | $FPR_{95}$ | AUC |
| ✗ | ✗ | 52.20 | 88.41 | 66.73 | 76.92 | 66.48 | 78.73 | 66.35 | 76.95 | 72.86 | 72.50 | 66.45 | 76.83 | 65.18 | 78.39 |
| ✓ | ✗ | 62.48 | 86.76 | 77.43 | 71.65 | 73.33 | 76.97 | 69.62 | 75.60 | 75.62 | 77.48 | 64.01 | 85.71 | 70.41 | 79.03 |
| ✗ | Jitter | 35.23 | 94.12 | 59.18 | 86.67 | 60.57 | 86.5 | 59.68 | 86.52 | 70.55 | 81.59 | 60.03 | 86.6 | 57.54 | 87.00 |
| ✓ | Jitter | 44.66 | 92.20 | 47.58 | 91.06 | 47.29 | 91.52 | 55.89 | 89.44 | 46.52 | 92.00 | 45.39 | 92.14 | 47.89 | 91.39 |

| MGP | | $FPR_{95}$ | AUC | $FPR_{95}$ | AUC | $FPR_{95}$ | AUC | $FPR_{95}$ | AUC | $FPR_{95}$ | AUC | $FPR_{95}$ | AUC | $FPR_{95}$ | AUC |
|---|---|---|---|---|---|---|---|---|---|---|---|---|---|---|---|
| ✗ | ✗ | **21.60** | **96.11** | 82.49 | 72.02 | 70.64 | 81.71 | 78.27 | 77.95 | 78.27 | 77.95 | 83.22 | 71.2 | 69.08 | 79.49 |
| ✓ | ✗ | 22.84 | 96.01 | 73.22 | 78.65 | 73.22 | 78.65 | 72.89 | 78.89 | 75.55 | 71.07 | 71.89 | 78.94 | 73.35 | 80.37 |
| ✗ | Jitter | 30.32 | 94.88 | 30.87 | 94.70 | 31.97 | 94.67 | 31.49 | 94.66 | 38.08 | 93.27 | 31.39 | 94.68 | 32.35 | 94.48 |
| ✓ | Jitter | 22.46 | 95.77 | **28.69** | **94.70** | **26.43** | **95.05** | **28.99** | **94.68** | **37.19** | **92.99** | **22.32** | **95.80** | **27.68** | **94.83** |

| SaGD | | Clean | | PGD | | Jitter | | FAB | | FGSM | | CW | | Average | |
|---|---|---|---|---|---|---|---|---|---|---|---|---|---|---|---|
| RSAM | AT | $FPR_{95}$ | AUC | $FPR_{95}$ | AUC | $FPR_{95}$ | AUC | $FPR_{95}$ | AUC | $FPR_{95}$ | AUC | $FPR_{95}$ | AUC | $FPR_{95}$ | AUC |
| ✓ | PGD | 86.64 | 60.31 | 95.62 | 75.42 | 94.54 | 60.95 | 88.25 | 77.74 | **20.25** | **95.19** | 86.60 | 60.36 | 78.65 | 71.66 |
| ✓ | FAB | 87.00 | 61.31 | 42.72 | 91.13 | 94.23 | 51.71 | 42.92 | 91.15 | 99.46 | 61.23 | 43.79 | 90.96 | 68.35 | 74.58 |
| ✓ | FGSM | 92.67 | 52.23 | 80.51 | 85.46 | 95.07 | 50.34 | 98.46 | 51.10 | 59.20 | 91.20 | 92.71 | 52.24 | 86.44 | 63.76 |
| ✓ | CW | 45.44 | 91.67 | 70.24 | 83.88 | 53.60 | 88.42 | 69.69 | 83.77 | 66.71 | 85.35 | 48.77 | 90.83 | 59.08 | 87.32 |

**Ablation study of geometry space, adversarial training, and RSAM:** The removal of the RSAM optimization module from our proposed SAGD adversely impacts both FPR95 and AUC. Specifically, MGP-Jitter experiences a decline in average $FPR_{95}$ to 32.35%, reflecting a 4.67% reduced margin compared to SaGD. Meanwhile, SaGD maintains a high average AUC of 94.48%, showing no significant decrease. Looking from another perspective, MGP-RSAM, discarding the Jitter adversarial training step from SaGD results in a significant increase of 45.67% in $FPR_{95}$ and a decrease of 14.46% in AUC. We also simplify the MGP structure as CIDER in SaGD which results in its combination with Jitter and RSAM. Jointly using Jitter and RSAM with CIDER obtains 47.89% $FPR_{95}$ which shows 22.58% and 9.65% improvements over CIDER-RSAM and CIDER-Jitter, respectively. Overall, the Jitter adversarial training benefits both CIDER and MGP frameworks. These results emphasize the significance of conducting Jitter adversarial training, and the RSAM approach can further facilitate the optimization steps.

**Evaluation on different adversarial training methods:** Based on the idea of generating adversarial examples for robust model training, we investigate additional adversarial attack approaches for adversarial training. Apart from Jitter, we incorporate PGD, FAB, FGSM, and CW to assess the OOD detection results under these different attacks. The lower part of Table 2 shows the average $FPR_{95}$ and AUC over six OOD testing datasets. Most adversarial attacks lead to substantial performance degradation. For example, PGD and FGSM share similar attack properties, resulting in average $FPR_{95}$ exceeding 80% for the model subjected to any attacks except FGSM. An intriguing result is observed with PGD, achieving 20.25% $FPR_{95}$ and a 95.19% AUC. This suggests that this type of perturbation can generate a notably robust model against the specific type of attack but may not generalize well to others.

## 4.3 Testing with Different Adversarial Paramters

**Perturbation intensity**: We investigate the influence of varying perturbation intensities ($\epsilon$) in the PGD adversarial attack on the SaGD method using the CIFAR-10 dataset. Figure 3 shows that $FPR_{95}$ is more susceptible to changes, while AUC maintains a consistently high standard as $\epsilon$ increases. Notably, an intense attack with $\epsilon = 16$ causes $FPR_{95}$ to double, whereas AUC experiences a 6.72% decline. These results suggest considering $FPR_{95}$ in the evaluation, an aspect that has been previously overlooked in the literature [1].

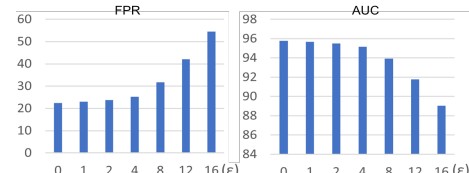

Figure 3: CIFAR-10 OOD detection results under different PGD attack perturbation intensities ($\epsilon$).

Table 3: Additional metrics (inlier and outlier AUC) and adaptive attacks on the CIFAR-10 dataset.

|  | \multicolumn{4}{c}{Average Adversarial} | \multicolumn{2}{c}{APGD-100} | \multicolumn{2}{c}{APGD-1000} | \multicolumn{2}{c}{Autoattack} |
|  | $FPR_{95}$ | AUC | $AUC_{In}$ | $AUC_{Out}$ | $FPR_{95}$ | AUC | $FPR_{95}$ | AUC | $FPR_{95}$ | AUC |
|---|---|---|---|---|---|---|---|---|---|---|
| ATD | 42.59 | 87.36 | 88.69 | 89.03 | 43.89 | 88.41 | 44.25 | 85.36 | 47.95 | 83.86 |
| SaGD | **27.68** | **94.83** | **95.71** | **95.86** | **28.67** | **94.18** | **29.13** | **94.50** | **32.18** | **93.01** |

**Iterative Attacks**: We include adaptive PGD (APGD) and autoattack [10] in Table 3. The adaptive PGD is investigated with 100 and 1000 steps and the autoattack is a parameter-free ensemble of multiple attacks. Increasing steps for APGD does not significantly affect the performance of ATD and SaGD compared to other types of attacks. The Autoattack obtains similar results with APGD in 1000 steps. SaGD can robustly defend for these different adversarial scenarios.

**Inlier AUC and Outlier AUC**: We analyze adversarial AUC metrics applying to inliers and outliers ($AUC_{In}$ and $AUC_{Out}$) which are also reported in [1]. Our targeted setting performing attacks on ID and OOD data results in lower AUC values than in $AUC_{In}$ and $AUC_{Out}$. SaGD can robustly achieve over 94% AUC and outperform ATD in the different metrics.

## 4.4 OOD Score Visualization

Figure 4 presents the OOD score histogram distribution between the CIFAR-10 ID testing data and TinyImageNet OOD testing data under clean and adversarial conditions. We demonstrate FGSM and FAB adversarial conditions. Other adversarial results with six OOD datasets are shown in the supplementary file. The ID data are colored in blue and the OOD data are in green. We consider models from the ablation study to further shed light on our proposed technical modules. Specifically, MGP, CIDER-RSAM-Jitter, and SaGD correspond to rows 5, 4, 9, and 8 in Table 2, respectively.

In clean conditions, MGP and SaGD distributions look alike, while CIDER-RSAM-Jitter shows a sharper OOD pattern. SaGD-PGD exhibits overlapping distributions between ID and OOD samples, albeit in a narrow area. Under the FGSM attack, MGP and CIDER-RSAM-Jitter distributions collapse significantly, blurring the line between ID and OOD samples. In contrast, SaGD maintains a consistent distribution, preserving a strong discriminative ability even under adversarial conditions. SaGD-PGD produces distinct peaks between ID and OOD distributions against PGD attacks. Investigating further, under FAB attacks, SaGD-PGD generates multiple peaks in the OOD distribution, confusing it with the long-tailed ID distribution. This highlights the overfitting challenges of adversarial training. These visualizations illustrate model properties regarding ID and OOD distributions, suggesting the potential of regularizing adversarial optimization across geometry spaces.

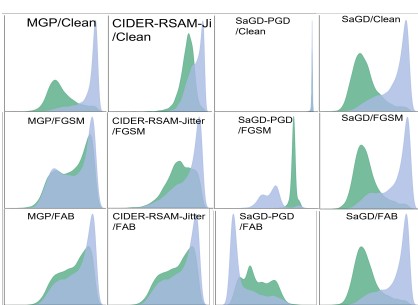

Figure 4: ID (blue) and OOD (green) score distribution in the clean condition, FGSM, and FAB adversarial conditions. We denote "detector/condition" where the detector can be MGP, CIDER-RSAM-Jitter, SaGD-PGD, or SaGD.

## 5 Conclusion

In this paper, we address the robustness issue for out-of-distribution (OOD) detection by investigating various types of adversarial attacks. We propose a novel SaGD framework that leverages the Jitter attack for adversarial training and optimizes the multi-geometry network using RSAM to enhance model convergence. The sharpness minimization strategy mitigates the rugged loss landscape induced by adversarial examples, resulting in improved OOD detection performance under attacks. Our OOD detection experiments encompass two in-distribution (ID) datasets and six OOD datasets tested against five types of attacks. SaGD achieves significantly low $FPR_{95}$ and high AUC on average. Our ablation study shows the critical role of Jitter-based adversarial training, highlighting the potential risk of employing popular perturbation approaches like PGD and FGSM. Our analysis shows the importance of using $FPR_{95}$ for evaluation as it tends to be impacted by increased attacks.

**Future work** includes the exploration of loss convergence conditions during adversarial geometry learning and improving the generalization of OOD detection capability under various adversarial conditions. We anticipate this work can initiate a novel direction to investigate an in-depth understanding of the relation between geometric loss optimization and robust OOD detection.

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

## A DETAILS OF ALGORITHMS

### A.1 HYPERBOLIC AVERAGE DERIVATION

The process of obtaining a hyperbolic average begins with the application of an *exponential map* to the embedding vector $\mathbf{v}$, transforming it to the tangent space on the Poincaré ball. This transformation expressed by the following equation essentially achieves hyperbolic embedding [25]:

$$\mathcal{E}^c(\mathbf{v}) = \tanh\left(\sqrt{c}||\mathbf{v}||\right)\frac{\mathbf{v}}{\sqrt{c}||\mathbf{v}||}. \tag{4}$$

The projected vectors in the hyperbolic space can use operations on Mobius gyrovector space with Mobius addition $\oplus_c$ and scalar multiplication $\otimes_c$, where $\mathbf{u}$ and $\mathbf{v}$ are vectors, and $w$ is a scalar.

$$\mathbf{u} \oplus_c \mathbf{v} = \frac{(1 + 2c < \mathbf{u}, \mathbf{v} > +c||\mathbf{v}||^2)\mathbf{u} + (1 - c||\mathbf{u}||^2)\mathbf{v}}{1 + 2c < \mathbf{u}, \mathbf{v} > +c^2||\mathbf{u}||^2||\mathbf{v}||^2},$$

$$w \otimes_c \mathbf{u} = \frac{1}{\sqrt{c}}\tanh\left(w \cdot \text{arctanh}\left(\sqrt{c}||\mathbf{u}||\right)\right)\frac{\mathbf{u}}{||\mathbf{u}||}. \tag{5}$$

Moving forward, we derive the process of *hyperbolic averaging* involving multiple hyperbolic embeddings through the Einstein midpoint. The embedding is projected from the Poincaré ball $\mathbb{D}_c^d$ to the Klein model $\mathbb{K}_c^d$, facilitating a simpler average computation in the Klein coordinate system:

$$\mathbf{u}_{\mathbb{K}} = \frac{2\mathbf{u}_{\mathbb{D}}}{1 + c||\mathbf{u}_{\mathbb{D}}||^2}, \quad \overline{\mathbf{u}_{\mathbb{K}}} = \frac{\sum_{i=1}^m r_i \mathbf{u}_{\mathbb{K},i}}{\sum_{i=1}^m r_i}, \tag{6}$$

where $r_i$ is the Lorentz factor.

Following the derivation of the average embedding within the Klein coordinate system, we then transform the space back to the Poincaré ball:

$$\overline{\mathbf{u}_{\mathbb{D}}} = \frac{\overline{\mathbf{u}_{\mathbb{K}}}}{1 + \sqrt{1 - c||\overline{\mathbf{u}_{\mathbb{K}}}||^2}}. \tag{7}$$

### A.2 CHARACTERISTICS OF VARIOUS ADVERSARIAL ATTACKS

In the main paper, we investigate a set of adversarial attacks to examine the robustness of OOD detection approaches. Since adversarial training is realized by using adversarial samples generated by adversarial attacks, we discuss the characteristics of these adversarial attacks in this section.

Given a data sample $x$ with label $y$, an adversarial sample $x^* = x + \gamma$ is generated to attack the target model $f_\theta$ in an optimization process aiming to maximize the following equation with a perturbation intensity $\gamma$ smaller than the upper-bound $\epsilon$:

$$\max_{||x-x^*||_\infty < \epsilon} \mathcal{L}(x^*, y; f_\theta), \tag{8}$$

where $\mathcal{L}$ is a targeted loss function, which is usually a cross-entropy for the targeted classification task. On the other hand, the adversarial training for defense is another optimization process aiming to determine the minimal impact from these adversarial samples, which can be written as:

$$\underset{\theta}{\text{argmin}}\ \mathbb{E}_{(x,y)\sim D_{in}}\ \underset{||x-x^*||_\infty<\epsilon}{\min}\ \mathcal{L}(x^*,y;f_\theta). \tag{9}$$

We next summarize popular adversarial attacks that are considered in the main paper.

**Fast Gradient Sign Method (FGSM)** [18]: The basic idea is to determine the gradient of the loss $\nabla\mathcal{L}$ in order to amplify the loss. The adversarial sample $x^*$ is formed by combining the original sample $x$ with a perturbation:

$$x^* = x + \epsilon\ \text{sign}\left(\nabla\mathcal{L}(x,y;f_\theta)\right). \tag{10}$$

**Projected Gradient Descent (PGD)** [34]: enhances the attack by using a $t$-step updating iteration on the greatest loss gradient with a step size $\eta$. The initial step starts with adding a random noise from a uniform distribution $\mathcal{U}(-\epsilon,\epsilon)$:

$$x^*_{t+1} = \underset{||x-x^*_t||_\infty<\epsilon}{\max}\ \left\{x^*_t + \eta\ \text{sign}\left(\nabla\mathcal{L}(x^*_t,y;f_\theta)\right)\right\} \tag{11}$$

The **Jitter** attack [41] is described in the main manuscript.

The **Fast Adaptive Boundary (FAB)** [9] attack focuses on making a correctly classified sample $x_0$ to be misclassified by finding the decision hyperplane close to $x_0$ and performing extrapolation. This hyperplane projection $\pi_s : \langle w, x\rangle + b$ with the parameters $w \in \mathbb{R}^d$ and $b \in \mathbb{R}$ under a box constrain $C = \{z \in \mathbb{R}^d : l_i \le z_i \le u_i, i = 1, 2, ..., d\}$ can be estimated to be the closest decision boundary of $x_0$ using first-order Taylor expansion. That is the projected point on the closest decision boundary, $z_0$ fulfills the optimization result:

$$z_0 = \underset{z}{\text{argmin}}||z - x_0||_p, \tag{12}$$

with $\langle w, z\rangle + b = 0$. A box-constrained hyperplane projection is then described as:

$$Proj_p(x, \pi_s, C) \rightarrow \begin{cases} z_0 & \text{if eq. equation 12 is hold} \\ z' & \text{else} \end{cases} \tag{13}$$

where $z'$ is another condition in the optimization iterations that $z$ is not sat on the hyperplane, with $\rho = \text{sign}(\langle w, z\rangle + b)$

$$z' = \begin{cases} l_i & \text{if } \rho w_i > 0 \\ u_i & \text{if } \rho w_i < 0 \\ x_i & \text{if } w_i = 0, \text{for } i = 1, 2, ..., d \end{cases} \tag{14}$$

When we obtain the projection to the closest hyperplane for $x_0$ as $Proj_p(x_0, \pi_s, C)$, we can perform extrapolation to derive the resulting adversarial sample $x^*$ which does not violate the box constrain.

The **CW** [3] attack is named after Carlini and Wagner, the inventors constructing adversarial samples in the following $\tanh$ space: $w^* =$

$$\min_w \left\|\frac{1}{2}(\tanh(w) + 1) - x\right\|_2^2 + c \cdot g\left(\frac{1}{2}(\tanh(w) + 1)\right), \tag{15}$$

$$x^* = \frac{1}{2}(\tanh(w^*) + 1), \tag{16}$$

where the hyperparameter $c$ determines the intensity of the perturbation and $g(x) = max\left(f(x)_y - max_{i\neq y}(f(x)_i), \kappa\right)$ indicates the aim to encourage $f(x)_y - max_{i\neq y}(f(x)_i)$ in proximity to $\kappa$. The CW attack modifies the variable in the $\tanh$ optimization space, with the underlying rationale of smoothing the clipped gradient to steer clear of local suboptimal points of $w^*$.

### A.3 LOSS LANDSCAPE OF ADVERSARIAL TRAINING

The practice of adversarial training serves as a countermeasure to defense against adversarial attacks. However, previous studies have uncovered challenges related to model convergence, attributed to the intricate loss landscape shaped by adversarial samples during training steps [51, 7]. We next delve into the theoretical underpinnings of this phenomenon.

The weight of a $j$-th layer in a neural network $f_\theta$ can be expressed as: $W^T = W_j^T \Sigma_{j-1} \ddot{W}_2^T \Sigma_2 W_1^T \Sigma_1$, which generetes gradient $g_W = \frac{\partial}{\partial W} L(x, y; f_\theta)$. Introducing adversarial samples $x^* = x + \gamma$, the corresponding gradient becomes $g_W^* = \frac{\partial}{\partial W} L(x + \gamma, y; f_\theta)$. Considering $\Delta g_W = g_W^* - g_W$ as the additional gradient values resulting from adversarial training, the full expression for gradient $\delta g_W$ is as follows:

$$\Delta g_W = \frac{\partial}{\partial W} L(x, y; f_\theta) - \frac{\partial}{\partial W} L(x + \gamma, y; f_\theta), \tag{17}$$

where $\gamma$ can be regularized by a $l_2$ or $l_\infty$ norm. This adversarial perturbation $\gamma$ is generated by $m$ steps of attacks with each step size $\alpha$, where $m$ should be large and $alpha$ is small.

we next analyze the case of binary classification, where the multi-class classification tasks can be simplified by focusing on the difference between the prediction for the targeted class $z_1'$ and the second highest probable class $z_2'$. Misclassification occurs when the probability of the second class surpasses that of the targeted class, and this difference is denoted as $z = z_1' - z_2' \in \mathbb{R}$. The effect of introducing adversarial samples brings in a change of the gradient $\tilde{g}_x = \frac{\partial z(x)}{\partial x}$. The additional gradient for updating the model with a learning rate $\eta$ in adversarial training is expressed as $\Delta \tilde{g}_x = -\eta \Delta g_W \tilde{g}_h$. Here, $\tilde{g}_h = \frac{\partial z(x)}{\partial h}$ indicates the gradient of the network output $z(x)$ with respect to the latent layer $h$.

Based on a lamma described in [7], we know that the following relations hold with $\mathcal{A} = m\alpha H_z ||\tilde{g}_x||^2 \in \mathbb{R}$:

$$H_x \Delta \tilde{g}_W = (e^{\mathcal{A}} - 1) H_x x \tilde{g}_h^T - \frac{1}{H_z ||\tilde{g}_x||^2} (e^{2\mathcal{A}} - e^{\mathcal{A}}) H_x g_x g_h^T. \tag{18}$$

The Hessian matrix is $H_h = \frac{\partial^2}{\partial h \partial h^T} L(x + \gamma, y; f_\theta)$ which can be rewritten as $H_h = H_z \tilde{g}_h \tilde{g}_h^T$ and $H_x = H_z \tilde{g}_x \tilde{g}_x^T$.

Therefore, we can assess the significance of this change $\Delta \tilde{g}_x$ along the direction of $\tilde{g}_x$:

$$\tilde{g}_x^T \Delta \tilde{g}_x = -\eta \tilde{g}_x^T \Delta g_W \tilde{g}_h \tag{19}$$

$$= (e^{\mathcal{A}} - 1) \tilde{g}_x^T \Delta \tilde{g}_x^0 - \frac{\eta g_z^2 ||\tilde{g}_h||^2}{H_z} (e^{2\mathcal{A}} - e^{\mathcal{A}}), \tag{20}$$

where $\Delta \tilde{g}_x^0 = -\eta g_W \tilde{g}_h$. Meanwhile, the significance measuring for the adversarial training along the direction of $\tilde{g}_x$ is as follows:

$$\tilde{g}_x^T \Delta \tilde{g}_x^* = -\eta \tilde{g}_x^T \Delta g_W^* \tilde{g}_h \tag{21}$$

$$= e^{\mathcal{A}} \tilde{g}_x^T \Delta \tilde{g}_x^0 - \frac{\eta g_z^2 e^{2\mathcal{A}} - e^{\mathcal{A}}}{H_z} ||\tilde{g}_h||^2. \tag{22}$$

This design of adversarial training expects the gradient $g_x$ with $g_x^T \Delta \tilde{g}_x < 0$. However, this assumption might not be held as the second term of equation equation 20 and equation equation 22 can be negative owing to $H_z > 0$. A few unconfident samples tend to generate large values for $H_z$ and large gradient values $||\tilde{g}_x||$. The phenomenon leads to difficulties in model convergence during adversarial training.

## B ADDITIONAL RESULTS WITH DETAILS

In our experiments, we assess six Out-of-Distribution (OOD) datasets in conjunction with various In-Distribution (ID) datasets. The OOD detection experiments involve diverse adversarial attacks, yielding a multitude of results. Therefore, the main paper incorporates averaged detection outcomes across the six OOD datasets. In this section, we provide a comprehensive breakdown of these results for each dataset, elucidating the distinctive characteristics under various adversarial conditions.

### B.1 EVALUATION OF OUT-OF-DISTRIBUTION ACCURACY

We report additional post-processing based OOD detection baselines including MaxSoftmax [21], Energy [33], MaxLogits [23], KLMatching [23], Entropy [4], Mahalanobis [29], MaxSoftmax [21], Energy [33], MaxLogits [23] and KLMatching [23], ODIN [31], ViM [48], GODIN [24], and ASH [12]. The OOD detection results for CIFAR-10 and CIFAR-100 datasets in the clean and adversarial conditions are demonstrated in Table 4 and Table 5, respectively. These OOD detection methods without adversarial training are vulnerable to adversarial attacks, yet these are all commonly used OOD detection methods. The results in Table 4 and Table 5 show that these OOD methods obtain over 80% $\text{FPR}_{95}$ and lower AUC than 60%. In contrast, MGP demonstrates potential robustness over adversarial attacks shown in Table 7 and Table 8.

We also report the detailed results using KNN+, SSD, CIDER, MGP, ATD, and SaGD on each OOD dataset based on CIFAR-10 as the ID dataset with their $\text{FPR}_{95}$ and AUC in Table 7. The results using the CIFAR-100 dataset as the ID dataset are presented in Table 8. For CIDER and MGP, we consider both KNN and Mahalanobis distance measurement approaches.

In differentiation the CIFAR-10 dataset from the other OOD dataset, KNN+ stands out in four of the six OOD datasets with 18.06% $\text{FPR}_{95}$ and 96.59% AUC averaged over the six datasets in the clean condition. SaGD keeps a high standard of $\text{FPR}_{95}$ and AUC even though it is trained for adversarial conditions. For the five attacks including PGD, Jitter, FAB, FGSM, and CW, Our proposed SaGD almost achieved the best performance on each dataset. The only exception occurs when using CIDER-Maha in the LSUN dataset. Nevertheless, CIDER-Maha attains minimal model performance on other datasets which causes nearly doubled $\text{FPR}_{95}$. The LSUN dataset is relatively separable from the CIFAR-10 datasets, allowing several models to achieve single-digit $\text{FPR}_{95}$ and over 95% AUC in both clean and adversarial scenarios.

In differentiation the CIFAR-100 dataset from the other OOD dataset, SaGD outperforms other methods in both adversarial and clean conditions. Given the difficulty of the OOD detection task posed by the CIFAR-100 dataset in comparison to CIFAR-10, SaGD shines in its remarkable robustness. This resilience is attributed to the effective smoothing of the loss landscape during adversarial training conditions. Introducing perturbations during training significantly boosts resilience in the challenging OOD task, particularly when CIFAR-100 is utilized as the ID dataset. Notably, MGP-Maha occasionally achieves low $\text{FPR}_{95}$ and high AUC when detecting data from LSUN, which is a relatively easy OOD dataset. This phenomenon is also observed with CIDER-Maha under FGSM attacks, albeit without significant impact on other datasets. We found that these OOD detction methods were substantially affected by numerous attacks on the iSUN and LSUN-R datasets. Examples include the extremely high $\text{FPR}_{95}$ (over 90%) using MGP-KNN under PGD, Jitter, FAB, and FGSM. Although ATD reduces the high $\text{FPR}_{95}$ on the LSUN-R and iSUN datasets, the 61.80% $\text{FPR}_{95}$ and 62.16% AUC under PGD attacks are still much worse than SaGD with 32.91% and 36.15% $\text{FPR}_{95}$ on the LSUN-R and iSUN datasets, respectively.

Across the six OOD datasets, SaGD exhibits comparable performance under the five adversarial conditions and the best results in clean sets. However, there is still room for improvement in its performance on TinyImgNet and Place365, presenting an opportunity for further enhancements in OOD detection methods.

### B.2 ABLATION STUDY

Table 9 presents a detailed ablation study conducted on CIFAR-10 in differentiation of the six other OOD datasets. The left section investigates OOD detection performance with and without Jitter adversarial training, as well as RSAM using hypersphere geometry (CIDER) or multiple-geometry (MGP) learning schemes. On the right, various adversarial training methods within the SaGD framework are explored. The results are elucidated in the following two sections.

Throughout the subsequent paragraphs, we adopt the format "Model-Adversarial Training-RSAM" to denote the components employed in the ablation study. Our proposed SaGD represents MGP-Jitter-RSAM, allowing for different combinations with various models and adversarial training methods. To describe the right part of Table 9, we substitute Jitter adversarial training with alternative methods, namely PGD, FAB, FGSM, and CW, which are denoted as SaGD-PGD, SaGD-FAB, SaGD-FGSM, SaGD-CW, respectively.

Table 4: Evaluation for different OOD detection methods **without adversarial training** using *CIFAR-10* as ID samples and the other six datasets as OOD samples. We report the average FPR$_{95}$ and AUC scores across the six tests.

| CIFAR10 | Clean FPR$_{95}$ | Clean AUC | PGD FPR$_{95}$ | PGD AUC | Jitter FPR$_{95}$ | Jitter AUC | FAB FPR$_{95}$ | FAB AUC | FGSM FPR$_{95}$ | FGSM AUC | CW FPR$_{95}$ | CW AUC | Average FPR$_{95}$ | Average AUC |
|---|---|---|---|---|---|---|---|---|---|---|---|---|---|---|
| KLMatching | 74.41 | 86.85 | 92.89 | 48.53 | 89.81 | 58.64 | 92.88 | 48.53 | 92.17 | 52.77 | 92.95 | 48.06 | 89.19 | 57.23 |
| MaxSoftmax | 35.89 | 90.25 | 95.16 | 47.25 | 90.19 | 57.98 | 95.16 | 47.25 | 93.72 | 52.67 | 96.05 | 46.49 | 84.36 | 56.98 |
| EnergyBased | 40.82 | 91.32 | 94.21 | 48.96 | 88.23 | 65.94 | 94.21 | 48.96 | 92.46 | 52.42 | 94.99 | 47.79 | 84.15 | 59.23 |
| MaxLogit | 40.88 | 91.28 | 94.19 | 48.90 | 88.37 | 65.52 | 94.19 | 48.90 | 92.55 | 52.48 | 95.03 | 47.81 | 84.20 | 59.15 |
| Entropy | 32.18 | 91.59 | 95.14 | 48.00 | 90.46 | 59.14 | 95.14 | 48.00 | 93.75 | 52.83 | 96.00 | 47.16 | 82.11 | 57.79 |
| ViM | 29.17 | 92.98 | 91.65 | 60.59 | 81.68 | 71.03 | 91.65 | 60.59 | 87.61 | 65.78 | 92.20 | 60.14 | 78.99 | 68.52 |
| Mahalanobis | 17.46 | 96.84 | 75.03 | 73.58 | 70.96 | 74.83 | 75.04 | 73.57 | 74.59 | 73.55 | 76.23 | 72.53 | 64.89 | 77.48 |
| ODIN | 42.51 | 91.12 | 92.93 | 55.22 | 89.71 | 63.30 | 92.93 | 55.22 | 91.38 | 61.47 | 94.68 | 53.98 | 84.02 | 63.39 |
| GODIN | 18.72 | 96.10 | 70.87 | 83.81 | 60.59 | 84.76 | 71.08 | 83.75 | 70.33 | 83.23 | 70.68 | 83.95 | 60.38 | 85.93 |
| ASH | 27.53 | 94.08 | 71.31 | 78.45 | 68.54 | 79.58 | 81.79 | 71.12 | 87.07 | 67.86 | 81.66 | 70.90 | 69.65 | 77.00 |
| KNN+ | 18.06 | 96.59 | 77.52 | 78.5 | 77.41 | 78.83 | 76.32 | 75.14 | 77.26 | 78.81 | 76.26 | 79.65 | 67.14 | 81.25 |
| SSD | 33.08 | 94.87 | 79.72 | 65.01 | 78.28 | 68.3 | 80.03 | 65.13 | 84.13 | 78.88 | 79.39 | 65.08 | 69.11 | 72.88 |
| CIDER-KNN | 52.20 | 88.41 | 66.73 | 76.92 | 66.48 | 78.73 | 66.35 | 76.95 | 72.86 | 72.50 | 66.45 | 76.83 | 65.18 | 78.39 |
| CIDER-Maha | 51.19 | 88.91 | 58.95 | 83.47 | 54.29 | 87.43 | 59.32 | 83.49 | 49.96 | 88.42 | 59.41 | 83.51 | 55.52 | 85.87 |
| MGP-KNN | **21.60** | **96.11** | 82.49 | 72.02 | 70.64 | 81.71 | 78.27 | 77.95 | 78.27 | 77.95 | 83.22 | 71.2 | 69.08 | 79.49 |
| MGP-Maha | 29.98 | 95.36 | **55.39** | **87.81** | **50.43** | **90.52** | **55.49** | **87.93** | **38.91** | **91.82** | **54.28** | **87.92** | **47.41** | **90.23** |

Table 5: Evaluation for different OOD detection methods **without adversarial training** using *CIFAR-100* as ID samples and the other six datasets as OOD samples. We report the average FPR$_{95}$ and AUC scores across the six tests.

| CIFAR100 | Clean FPR$_{95}$ | Clean AUC | PGD FPR$_{95}$ | PGD AUC | Jitter FPR$_{95}$ | Jitter AUC | FAB FPR$_{95}$ | FAB AUC | FGSM FPR$_{95}$ | FGSM AUC | CW FPR$_{95}$ | CW AUC | Average FPR$_{95}$ | Average AUC |
|---|---|---|---|---|---|---|---|---|---|---|---|---|---|---|
| KLMatching | 94.57 | 44.52 | 95.15 | 54.24 | 93.79 | 58.54 | 90.67 | 56.51 | 93.88 | 51.79 | 90.40 | 55.57 | 93.08 | 53.53 |
| MaxSoftmax | 88.78 | 58.99 | 92.39 | 53.10 | 87.91 | 59.77 | 79.35 | 61.98 | 93.17 | 49.65 | 92.42 | 52.97 | 89.00 | 56.08 |
| EnergyBased | 83.97 | 63.75 | 89.46 | 55.95 | 81.47 | 65.41 | 74.47 | 61.83 | 89.85 | 52.23 | 89.37 | 55.53 | 84.72 | 59.12 |
| MaxLogit | 84.35 | 63.45 | 89.63 | 55.19 | 81.88 | 64.96 | 74.47 | 62.02 | 90.29 | 51.46 | 89.52 | 55.26 | 85.02 | 58.72 |
| Entropy | 88.62 | 60.42 | 92.36 | 53.65 | 87.96 | 61.67 | 79.30 | 62.14 | 93.03 | 49.89 | 92.11 | 53.88 | 88.90 | 56.94 |
| ViM | 75.94 | 73.34 | 83.55 | 63.17 | 80.16 | 65.52 | 74.00 | 69.21 | 84.34 | 58.72 | 80.10 | 68.91 | 79.68 | 66.48 |
| Mahalanobis | 72.21 | 74.22 | 81.38 | 63.00 | 81.53 | 63.07 | 77.23 | 66.09 | 82.79 | 59.45 | 75.42 | 69.63 | 78.43 | 65.91 |
| ODIN | 81.57 | 68.05 | 89.86 | 58.54 | 84.56 | 63.94 | 76.88 | 62.07 | 91.18 | 54.42 | 89.96 | 58.33 | 85.67 | 60.89 |
| GODIN | 74.58 | 80.88 | 90.19 | 67.37 | 90.45 | 73.03 | 94.66 | 66.41 | 95.33 | 64.85 | 90.39 | 67.39 | 89.27 | 69.99 |
| ASH | 59.04 | 84.44 | 73.74 | 62.56 | 70.37 | 77.93 | 78.18 | 70.14 | 85.89 | 65.01 | 75.24 | 75.72 | 73.69 | 72.63 |
| KNN+ | 65.47 | 85.07 | 95.59 | 51.07 | 95.39 | 51.38 | 95.44 | 51.61 | 95.55 | 50.53 | 95.38 | 51.58 | 90.47 | 56.87 |
| SSD | 70.98 | 84.94 | 95.16 | 46.80 | 94.51 | 49.60 | 95.28 | 46.75 | | | 95.10 | 46.60 | 90.56 | 54.53 |
| CIDER-KNN | 65.99 | 83.44 | 75.69 | **82.95** | 66.59 | 82.07 | **74.70** | **83.08** | 78.24 | 77.43 | 76.10 | 83.06 | 72.88 | **82.01** |
| CIDER-Maha | 67.28 | 84.36 | **75.47** | 75.85 | **63.36** | **83.94** | 74.83 | 75.76 | **53.26** | **84.38** | 76.17 | 75.91 | **68.40** | 80.03 |
| MGP-KNN | **57.89** | **85.26** | 80.23 | 76.17 | 81.41 | 74.63 | 79.8 | 76.28 | 78.74 | 69.99 | **60.05** | **82.79** | 73.02 | 77.52 |
| MGP-Maha | 66.47 | 83.19 | 79.33 | 65.81 | 74.12 | 74.96 | 81.32 | 62.72 | 64.99 | 75.98 | 76.60 | 73.09 | 73.80 | 72.62 |

### B.2.1 ABLATION STUDY OF GEOMETRY SPACE, ADVERSARIAL TRAINING, AND RSAM

In Table 9, the removal of multi-geometry learning, Jitter, or RSAM individually leads to a decline in FPR$_{95}$ and AUC. Comparing CIDER and MGP with RSAM, MGP maintains a low FPR$_{95}$ in the clean condition, while both methods exhibit high FPR$_{95}$ in adversarial conditions, hovering around 70%. CIDER and MGP with Jitter adversarial training notably enhance results compared to models using RSAM. The clean condition FPR$_{95}$ for CIDAR-Jitter outperforms CIDER-RSAM by 27.25%, with improvements across all datasets except LSUN, where the FPR$_{95}$ is already low. MGP-Jitter achieves low FPR$_{95}$ in both clean and adversarial conditions.

The lowest averaged FPR$_{95}$ occurs in PGD, and the highest averaged FPR$_{95}$ is 38.08% in FGSM, close to the 30.32% averaged FPR$_{95}$ in clean condition. Among the six OOD datasets using MGP-Jitter, the TinyImageNet dataset poses a challenging task with 54.57% FPR$_{95}$, while the LSUN dataset achieves 18.40% FPR$_{95}$ under FGSM attacks.

CIDER-RSAM-Jitter improves upon CIDER-RSAM and CIDER-Jitter, achieving a notable 7.40% FPR$_{95}$ in the LSUN dataset under Jitter attacks. Moreover, CIDER-RSAM-Jitter demonstrates significant improvement with a 22.67% FPR$_{95}$ in the Places356 dataset under FGSM attacks compared to the result of CIDER-Jitter. Despite the effectiveness of using Jitter adversarial training and RSAM validated by CIDER-RSAM-Jitter, our proposed SaGD consistently outperforms other methods with better averaged FPR$_{95}$ and AUC across all OOD datasets.

Table 6: The average $FPR_{95}$ and AUC scores **with adversarial training** over six datasets using *CIFAR-100* as ID samples. *Clean* denotes the standard OOD detection without any attack. We consider the OOD detection under *PGD, Jitter, FAB, FGSM, CW* attacks.

| CIFAR10 | Clean | | PGD | | Jitter | | FAB | | FGSM | | CW | | Average | |
|---|---|---|---|---|---|---|---|---|---|---|---|---|---|---|
| | $FPR_{95}$ | AUC | $FPR_{95}$ | AUC | $FPR_{95}$ | AUC | $FPR_{95}$ | AUC | $FPR_{95}$ | AUC | $FPR_{95}$ | AUC | $FPR_{95}$ | AUC |
| ATD | 27.46 | 94.15 | 57.04 | 79.63 | 43.61 | 87.60 | 33.09 | 92.38 | 37.56 | 90.70 | 56.78 | 79.69 | 42.59 | 87.36 |
| SaMD | **22.46** | **95.77** | **28.69** | **94.70** | **26.43** | **95.05** | **28.99** | **94.68** | **37.19** | **92.99** | **22.32** | **95.80** | **27.68** | **94.83** |

| CIFAR100 | Clean | | PGD | | Jitter | | FAB | | FGSM | | CW | | Average | |
|---|---|---|---|---|---|---|---|---|---|---|---|---|---|---|
| | $FPR_{95}\downarrow$ | AUC$\uparrow$ | $FPR_{95}\downarrow$ | AUC$\uparrow$ | $FPR_{95}\downarrow$ | AUC$\uparrow$ | $FPR_{95}\downarrow$ | AUC$\uparrow$ | $FPR_{95}\downarrow$ | AUC$\uparrow$ | $FPR_{95}\downarrow$ | AUC$\uparrow$ | $FPR_{95}\downarrow$ | AUC$\uparrow$ |
| ATD | 63.04 | 82.90 | 73.79 | 66.18 | 64.90 | 80.07 | 63.04 | 82.90 | 70.38 | 76.37 | 70.30 | 76.06 | 67.58 | 77.41 |
| SaMD | **50.89** | **87.26** | **48.31** | **88.13** | **47.63** | **88.15** | **49.08** | **87.88** | **52.51** | **86.78** | **50.81** | **87.32** | **49.87** | **87.59** |

### B.2.2 EVALUATION ON DIFFERENT ADVERSARIAL TRAINING METHODS

In the right section of Table 9, we substitute Jitter adversarial attacks with alternative approaches for adversarial training. The results are discussed as follows.

Utilizing FGSM, PGD, and FAB for training significantly increases $FPR_{95}$ in the clean condition, with the worst $FPR_{95}$ reaching 97.17%, 99.94%, and 95.46% for the three adversarial training methods. These adversarial training approaches are not robust enough to defend against different attacks, leading to unexpectedly high $FPR_{95}$. Models trained with FGSM or PGD exhibit over 80% averaged $FPR_{95}$ under attacks, except for FGSM. Although training with FAB successfully defends against PGD, FAB, and CW during testing, it fails to perform OOD detection under Jitter and FGSM attacks. Notably, the failure of defense tends to occur simultaneously across all tested OOD datasets rather than being specific to a single dataset.

Training with CW yields more consistent defensive results, with no attack showing a clear preference for using CW over Jitter. Although CW adversarial training reaches 11.02% and 32.67% $FPR_{95}$ in the LSUN and Textures datasets, respectively, SaGD-Jitter achieves better results with 3.91% and 17.87% $FPR_{95}$ in the same datasets.

### B.3 OOD SCORE VISUALIZATION

We analyze the OOD detection results by plotting the histogram of the OOD score for both ID and OOD data in blue and green, as shown in Figure 5. The histograms are represented in blue and green, respectively, in the subfigure comparing MGP, SaGD, CIDER-Jitter-RSAM, and SaGD-PGD.

Generally, SaGD exhibits the best performance, leading to more separable distributions, while the other methods have more overlapped regions. MGP, on the other hand, displays varying distribution plots between the clean condition and other adversarial conditions. In contrast, SaGD and CIDER-Jitter-RSAM consistently exhibit similar distribution plots across different conditions, indicating minimal influence from the applied attacks.

Through this visualization analysis, we glean insights into the model's robustness. For example, SaGD and CIDER-Jitter-RSAM display smoother visualization results, whereas MGP generates additional peaks. Despite a tail persisting in the ID distribution of SaGD, it remains distant from the majority of ID samples. On the other hand, CIDER-Jitter-RSAM sometimes yields a slim ID distribution but significantly overlaps with the OOD data, as evident in the Textures dataset.

In the ablation study, considering PGD as a substitute for Jitter adversarial training reveals defense failures under multiple attacks. SaGD-PGD introduces additional spikes in FAB attacks, resulting in a high and sharp peak in both ID and OOD distributions. However, these two distributions exhibit significant overlap, indicating poorly converged results. Our proposed SaGD, designed to address the non-smooth loss landscape in adversarial training, consistently manifests smoother OOD score distributions.

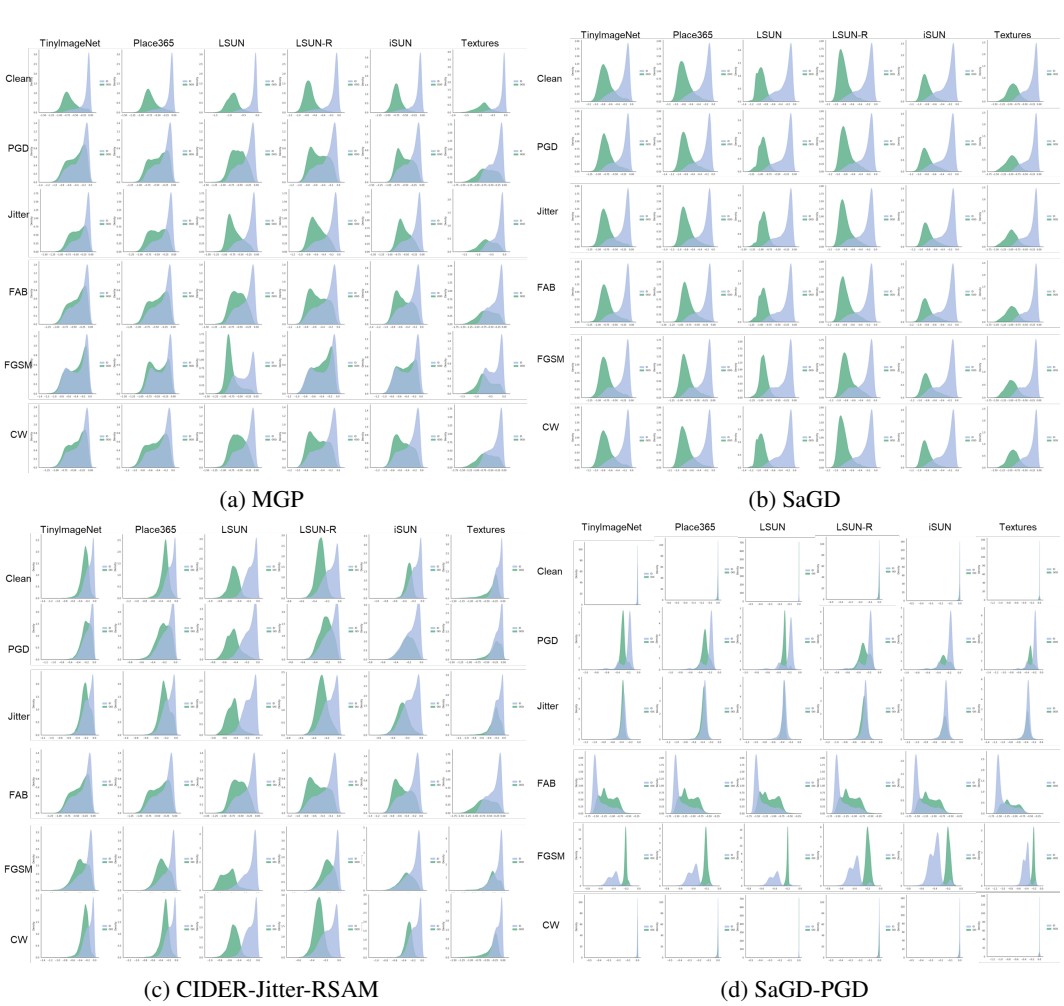

(a) MGP

(b) SaGD

(c) CIDER-Jitter-RSAM

(d) SaGD-PGD

Figure 5: Complete visualization of the OOD score histograms conducted using CIFAR-10 as ID dataset for the comparison of (a) MGP, (b) SaGD, (c) CIDER-Jitter-RSAM, and (d) SaGD-PGD methods. The ID and OOD samples are colored in blue and green, respectively. Each column represents results from an OOD dataset, and each row indicates different adversarial conditions.

Table 7: Full evaluation results for different OOD detection methods using *CIFAR-10* as ID samples and the other six datasets as OOD samples.

| | KNN+ FPR$_{95}$ | KNN+ AUC | SSD FPR$_{95}$ | SSD AUC | CIDER FPR$_{95}$ | CIDER AUC | CIDER-Maha FPR$_{95}$ | CIDER-Maha AUC | MGP FPR$_{95}$ | MGP AUC | MGP-Maha FPR$_{95}$ | MGP-Maha AUC | ATD FPR$_{95}$ | ATD AUC | SaGD FPR$_{95}$ | SaGD AUC |
|---|---|---|---|---|---|---|---|---|---|---|---|---|---|---|---|---|
| **Clean** | | | | | | | | | | | | | | | | |
| TinyImgNet | 31.47 | 93.21 | 34.10 | 92.70 | 69.19 | 84.31 | 65.29 | 85.02 | 36.89 | 92.64 | 36.08 | 92.81 | 45.27 | 88.42 | 41.31 | 91.75 |
| Place365 | 21.65 | 95.56 | 23.58 | 95.33 | 66.75 | 85.18 | 63.11 | 85.92 | 29.61 | 94.52 | 31.36 | 94.44 | 36.64 | 92.89 | 30.92 | 93.93 |
| LSUN | 1.23 | 99.62 | 2.11 | 99.49 | 4.14 | 99.15 | 3.43 | 98.91 | 9.70 | 98.35 | 13.25 | 97.98 | 24.40 | 94.52 | 3.93 | 99.09 |
| LSUN-R | 21.37 | 96.46 | 62.93 | 92.06 | 64.79 | 85.53 | 72.19 | 85.05 | 19.09 | 96.71 | 39.68 | 94.88 | 18.40 | 96.35 | 19.97 | 96.34 |
| iSUN | 24.81 | 96.05 | 67.23 | 91.12 | 61.38 | 85.81 | 67.98 | 85.57 | 18.92 | 96.78 | 45.30 | 94.26 | 26.60 | 95.04 | 20.71 | 96.38 |
| Textures | 7.82 | 98.63 | 8.51 | 98.50 | 46.93 | 90.50 | 35.14 | 93.01 | 15.39 | 97.66 | 14.18 | 97.79 | 13.45 | 97.70 | 17.89 | 97.14 |
| Average | 18.06 | 96.59 | 33.08 | 94.87 | 52.20 | 88.41 | 51.19 | 88.91 | 21.60 | 96.11 | 29.98 | 95.36 | 27.46 | 94.15 | 22.46 | 95.77 |
| **PGD** | | | | | | | | | | | | | | | | |
| TinyImgNet | 82.49 | 71.54 | 77.56 | 73.87 | 84.00 | 67.79 | 70.97 | 79.33 | 90.98 | 62.33 | 56.67 | 85.91 | 68.66 | 71.17 | 43.48 | 91.51 |
| Place365 | 81.80 | 73.37 | 79.71 | 74.49 | 78.00 | 73.68 | 67.72 | 81.57 | 86.14 | 66.8 | 50.97 | 89.5 | 59.20 | 79.48 | 34.65 | 93.15 |
| LSUN | 56.99 | 87.15 | 54.86 | 85.33 | 20.24 | 94.01 | 1.6 | 99.61 | 81.34 | 77.16 | 22.67 | 96.14 | 64.60 | 73.72 | 8.27 | 98.39 |
| LSUN-R | 98.87 | 63.36 | 99.71 | 39.86 | 71.67 | 78.66 | 86.41 | 75.04 | 85.65 | 73.19 | 79.31 | 83.37 | 47.24 | 85.44 | 34.99 | 93.86 |
| iSUN | 98.02 | 64.4 | 99.25 | 39.92 | 72.44 | 77.39 | 84.15 | 75.7 | 86.06 | 73.87 | 82.21 | 80.75 | 54.30 | 83.35 | 34.83 | 93.9 |
| Textures | 65.21 | 75.46 | 67.22 | 76.61 | 74.01 | 69.99 | 42.85 | 89.59 | 64.75 | 78.75 | 40.50 | 91.22 | 48.25 | 84.60 | 15.94 | 97.41 |
| Average | 80.56 | 72.55 | 79.72 | 65.01 | 66.73 | 76.92 | 58.95 | 83.47 | 82.49 | 72.02 | 55.39 | 87.81 | 57.04 | 79.63 | 28.69 | 94.70 |
| **Jitter** | | | | | | | | | | | | | | | | |
| TinyImgNet | 83.43 | 72.20 | 75.64 | 76.28 | 84.58 | 70.72 | 65.87 | 83.69 | 85.07 | 71.99 | 50.67 | 88.60 | 57.91 | 79.97 | 43.20 | 91.40 |
| Place365 | 82.93 | 73.97 | 76.99 | 77.75 | 81.16 | 72.35 | 62.71 | 85.65 | 78.74 | 76.87 | 44.22 | 91.44 | | | 2.79 | 99.52 |
| LSUN | 52.76 | 89.23 | 54.30 | 86.91 | 16.21 | 96.47 | 1.40 | 99.62 | 46.89 | 89.63 | 21.66 | 96.64 | 51.71 | 83.39 | 36.39 | 92.93 |
| LSUN-R | 98.91 | 66.02 | 99.63 | 44.53 | 72.81 | 79.22 | 80.43 | 81.65 | 75.98 | 83.63 | 72.57 | 87.62 | 34.06 | 91.46 | 8.90 | 98.28 |
| iSUN | 97.90 | 67.14 | 99.08 | 44.61 | 72.03 | 78.01 | 77.78 | 81.93 | 77.78 | 83.41 | 77.31 | 85.49 | 38.42 | 90.44 | 33.54 | 94.06 |
| Textures | 67.16 | 76.09 | 64.06 | 79.72 | 72.09 | 75.64 | 37.54 | 92.04 | 59.40 | 84.74 | 36.12 | 93.30 | 34.69 | 92.52 | 33.76 | 94.08 |
| Average | 80.51 | 74.11 | 78.28 | 68.30 | 66.48 | 78.73 | 54.29 | 87.43 | 70.64 | 81.71 | 50.43 | 90.52 | 43.61 | 87.60 | 26.43 | 95.05 |
| **FAB** | | | | | | | | | | | | | | | | |
| TinyImgNet | 55.11 | 85.21 | 77.72 | 74.05 | 83.57 | 68.04 | 71.18 | 79.78 | 88.19 | 69.77 | 56.53 | 85.84 | 68.65 | 71.17 | 43.64 | 91.46 |
| Place365 | 83.35 | 73.02 | 80.24 | 74.62 | 77.95 | 73.68 | 68.47 | 81.43 | 83.57 | 73.46 | 51.73 | 89.43 | 35.69 | 93.41 | 35.17 | 93.21 |
| LSUN | 55.33 | 87.41 | 55.36 | 85.41 | 19.68 | 93.99 | 1.62 | 99.59 | 76.13 | 82.52 | 22.81 | 96.20 | 49.03 | 85.59 | 8.38 | 98.36 |
| LSUN-R | 98.76 | 64.6 | 99.72 | 39.97 | 71.89 | 78.56 | 86.92 | 74.94 | 80.03 | 79.11 | 79.22 | 83.70 | 30.19 | 93.31 | 35.78 | 93.8 |
| iSUN | 98.2 | 65.19 | 99.28 | 40.04 | 71.66 | 77.23 | 84.5 | 75.6 | 80.74 | 79.55 | 82.16 | 81.08 | 34.29 | 92.60 | 34.55 | 93.86 |
| Textures | 67.16 | 75.44 | 67.87 | 76.70 | 73.33 | 70.21 | 43.21 | 89.58 | 60.99 | 83.27 | 40.51 | 91.31 | 16.27 | 96.98 | 16.4 | 97.41 |
| Average | 76.32 | 75.14 | 80.03 | 65.13 | 66.35 | 76.95 | 59.32 | 83.49 | 78.27 | 77.95 | 55.49 | 87.93 | 39.02 | 88.85 | 28.99 | 94.68 |
| **FGSM** | | | | | | | | | | | | | | | | |
| TinyImgNet | 81.33 | 76.86 | 52.98 | 86.16 | 88.78 | 63.96 | 52.47 | 87.66 | 93.10 | 54.20 | 36.43 | 91.07 | 59.96 | 82.31 | 50.51 | 90.12 |
| Place365 | 80.07 | 78.00 | 50.77 | 88.47 | 81.98 | 69.96 | 46.33 | 89.93 | 87.88 | 62.91 | 24.46 | 94.79 | 35.69 | 93.41 | 40.63 | 91.84 |
| LSUN | 44.14 | 92.28 | 33.22 | 94.13 | 20.01 | 94.85 | 0.79 | 99.8 | 44.31 | 88.49 | 11.79 | 97.97 | 49.03 | 85.59 | 14.18 | 97.54 |
| LSUN-R | 98.84 | 71.46 | 96.61 | 61.29 | 86.8 | 66.83 | 81.67 | 81.24 | 95.41 | 56.40 | 58.51 | 89.34 | 30.19 | 93.31 | 50.06 | 90.56 |
| iSUN | 97.79 | 72.14 | 95.85 | 59.86 | 85.22 | 66.88 | 80.82 | 80.59 | 94.05 | 59.40 | 65.04 | 86.42 | 34.20 | 92.60 | 47.27 | 91.2 |
| Textures | 61.37 | 82.11 | 55.37 | 83.36 | 74.38 | 72.50 | 37.71 | 91.28 | 65.57 | 76.37 | 37.25 | 91.30 | 16.27 | 96.98 | 20.48 | 96.67 |
| Average | 77.26 | 78.81 | 64.13 | 78.88 | 72.86 | 72.50 | 49.96 | 88.42 | 80.05 | 66.30 | 38.91 | 91.82 | 37.56 | 90.70 | 37.19 | 92.99 |
| **CW** | | | | | | | | | | | | | | | | |
| TinyImgNet | 82.63 | 70.92 | 76.53 | 74.08 | 83.66 | 67.85 | 71.34 | 79.70 | 91.39 | 61.72 | 54.67 | 86.14 | 68.69 | 71.17 | 41.3 | 91.78 |
| Place365 | 83.40 | 72.69 | 79.20 | 74.31 | 78.51 | 73.43 | 68.61 | 81.76 | 87.20 | 65.32 | 49.26 | 89.53 | 59.04 | 79.48 | 30.17 | 94.02 |
| LSUN | 57.4 | 86.97 | 54.53 | 85.39 | 20.16 | 93.92 | 1.71 | 99.58 | 81.57 | 76.54 | 22.07 | 96.19 | 64.48 | 73.95 | 3.91 | 99.09 |
| LSUN-R | 98.91 | 62.84 | 99.70 | 39.98 | 71.53 | 78.34 | 86.97 | 74.91 | 86.65 | 72.48 | 78.41 | 83.50 | 46.93 | 85.39 | 19.95 | 96.35 |
| iSUN | 97.86 | 63.82 | 99.25 | 40.05 | 71.27 | 77.18 | 84.62 | 75.56 | 86.33 | 73.18 | 81.38 | 80.88 | 53.40 | 83.49 | 20.71 | 96.39 |
| Textures | 65.35 | 75.01 | 67.16 | 76.66 | 73.55 | 70.28 | 43.24 | 89.52 | 66.19 | 77.95 | 39.91 | 91.27 | 48.13 | 84.63 | 17.87 | 97.14 |
| Average | 80.93 | 72.04 | 79.39 | 65.08 | 66.45 | 76.83 | 59.41 | 83.51 | 83.22 | 71.20 | 54.28 | 87.92 | 56.78 | 79.69 | 22.32 | 95.80 |

Table 8: Full evaluation results for different OOD detection methods using *CIFAR-100* as ID samples and the other six datasets as OOD samples.

| | KNN+ | | SSD | | CIDER-KNN | | CIDER-Maha | | MGP-KNN | | MGP-Maha | | ATD | | SaGD | |
|---|---|---|---|---|---|---|---|---|---|---|---|---|---|---|---|---|
| **Clean** | FPR$_{95}$ | AUC | FPR$_{95}$ | AUC | FPR$_{95}$ | AUC | FPR$_{95}$ | AUC | FPR$_{95}$ | AUC | FPR$_{95}$ | AUC | FPR$_{95}$ | AUC | FPR$_{95}$ | AUC |
| TinyImgNet | 76.18 | 80.21 | 77.92 | 79.97 | 76.33 | 79.23 | 76.43 | 80.52 | 74.06 | 81.49 | 78.49 | 79.87 | 78.38 | 74.71 | 76.70 | 79.82 |
| Place365 | 81.46 | 77.49 | 81.16 | 77.71 | 82.44 | 74.10 | 81.50 | 77.45 | 74.30 | 78.99 | 77.21 | 79.51 | 66.77 | 84.82 | 79.00 | 76.81 |
| LSUN | 49.30 | 90.10 | 41.00 | 92.78 | 43.31 | 89.72 | 24.70 | 95.29 | 11.87 | 96.94 | 21.09 | 96.15 | 79.13 | 72.81 | 38.37 | 91.06 |
| LSUN-R | 78.76 | 81.47 | 90.08 | 80.12 | 68.95 | 84.57 | 79.93 | 83.03 | 69.17 | 83.79 | 77.36 | 80.55 | 43.95 | 89.97 | 35.89 | 92.47 |
| iSUN | 80.08 | 80.20 | 90.61 | 78.93 | 68.21 | 84.31 | 80.85 | 82.33 | 69.99 | 82.06 | 80.64 | 78.82 | 56.02 | 87.50 | 38.90 | 91.09 |
| Textures | 55.90 | 89.15 | 60.14 | 88.05 | 56.67 | 88.73 | 60.25 | 87.56 | 47.93 | 88.28 | 64.01 | 84.25 | 53.97 | 87.56 | 36.45 | 92.33 |
| Average | 70.28 | 83.19 | 73.48 | 82.93 | 65.99 | 83.44 | 67.28 | 84.36 | 57.89 | 85.26 | 66.47 | 83.19 | 63.04 | 82.90 | 50.89 | 87.26 |
| **PGD** | FPR$_{95}$ | AUC | FPR$_{95}$ | AUC | FPR$_{95}$ | AUC | FPR$_{95}$ | AUC | FPR$_{95}$ | AUC | FPR$_{95}$ | AUC | FPR$_{95}$ | AUC | FPR$_{95}$ | AUC |
| TinyImgNet | 94.57 | 48.63 | 94.42 | 48.52 | 81.17 | 80.17 | 76.93 | 75.87 | 87.75 | 72.99 | 75.71 | 80.69 | 96.23 | 26.56 | 74.16 | 80.63 |
| Place365 | 95.18 | 49.15 | 95.62 | 47.76 | 83.57 | 78.38 | 76.90 | 76.02 | 88.35 | 72.29 | 88.63 | 67.34 | 66.97 | 79.48 | 76.19 | 78.30 |
| LSUN | 99.83 | 55.36 | 99.92 | 42.65 | 58.49 | 88.4 | 38.42 | 90.17 | 47.17 | 88.49 | 30.53 | 92.60 | 87.64 | 59.92 | 35.62 | 92.25 |
| LSUN-R | 99.59 | 49.92 | 99.78 | 40.28 | 78.42 | 82.74 | 89.94 | 71.97 | 92.35 | 72.95 | 94.96 | 52.70 | 62.16 | 77.30 | 32.91 | 93.05 |
| iSUN | 98.83 | 52.23 | 99.56 | 42.23 | 76.62 | 82.94 | 92.04 | 68.67 | 92.47 | 71.56 | 96.69 | 48.47 | 62.16 | 77.30 | 36.15 | 91.80 |
| Textures | 85.53 | 54.91 | 81.68 | 59.37 | 75.85 | 85.10 | 78.56 | 72.4 | 73.30 | 78.77 | 89.45 | 53.08 | 67.94 | 77.57 | 34.84 | 92.74 |
| Average | 95.59 | 51.70 | 95.16 | 46.80 | 75.69 | 82.95 | 75.47 | 75.85 | 80.23 | 76.18 | 79.33 | 65.81 | 73.79 | 66.18 | 48.31 | 88.13 |
| **Jitter** | FPR$_{95}$ | AUC | FPR$_{95}$ | AUC | FPR$_{95}$ | AUC | FPR$_{95}$ | AUC | FPR$_{95}$ | AUC | FPR$_{95}$ | AUC | FPR$_{95}$ | AUC | FPR$_{95}$ | AUC |
| TinyImgNet | 94.86 | 48.45 | 93.71 | 50.17 | 77.89 | 77.29 | 64.48 | 82.81 | 88.14 | 71.71 | 75.38 | 80.64 | 78.93 | 71.56 | 73.30 | 80.67 |
| Place365 | 95.10 | 48.79 | 94.83 | 49.44 | 80.00 | 75.21 | 65.89 | 81.96 | 88.27 | 69.67 | 82.86 | 74.11 | 64.66 | 82.54 | 75.98 | 78.22 |
| LSUN | 99.86 | 55.28 | 99.90 | 47.08 | 52.16 | 85.83 | 27.77 | 94.09 | 54.73 | 84.93 | 20.15 | 96.01 | 78.52 | 72.39 | 35.17 | 92.27 |
| LSUN-R | 99.63 | 48.35 | 99.69 | 43.88 | 64.70 | 82.94 | 75.76 | 82.26 | 91.48 | 71.62 | 90.80 | 66.69 | 48.62 | 86.76 | 31.64 | 93.15 |
| iSUN | 99.11 | 50.50 | 99.50 | 45.72 | 62.98 | 83.94 | 80.09 | 79.80 | 91.81 | 70.41 | 94.02 | 62.90 | 55.90 | 85.15 | 34.99 | 91.93 |
| Textures | 83.79 | 56.89 | 79.45 | 61.32 | 61.83 | 87.19 | 66.17 | 82.74 | 74.02 | 79.45 | 81.52 | 69.38 | 62.95 | 82.00 | 34.68 | 92.68 |
| Average | 95.39 | 51.38 | 94.51 | 49.60 | 66.59 | 82.07 | 63.36 | 83.94 | 81.41 | 74.63 | 74.12 | 74.96 | 64.93 | 80.07 | 47.63 | 88.15 |
| **FAB** | FPR$_{95}$ | AUC | FPR$_{95}$ | AUC | FPR$_{95}$ | AUC | FPR$_{95}$ | AUC | FPR$_{95}$ | AUC | FPR$_{95}$ | AUC | FPR$_{95}$ | AUC | FPR$_{95}$ | AUC |
| TinyImgNet | 94.55 | 48.87 | 94.43 | 48.57 | 79.98 | 80.51 | 75.73 | 75.72 | 87.36 | 72.96 | 90.55 | 60.77 | 78.38 | 74.71 | 75.75 | 79.69 |
| Place365 | 95.32 | 48.39 | 95.67 | 47.42 | 83.24 | 78.19 | 76.45 | 76.00 | 87.48 | 72.77 | 87.86 | 67.60 | 66.77 | 84.82 | 76.89 | 77.82 |
| LSUN | 99.70 | 55.40 | 99.95 | 42.70 | 57.43 | 88.60 | 37.85 | 90.22 | 46.80 | 88.47 | 29.46 | 92.78 | 79.13 | 72.81 | 36.22 | 92.24 |
| LSUN-R | 99.52 | 49.92 | 99.79 | 40.31 | 77.50 | 83.10 | 89.34 | 71.85 | 92.08 | 73.41 | 94.51 | 53.08 | 43.95 | 89.97 | 33.44 | 93.01 |
| iSUN | 98.98 | 52.12 | 99.62 | 42.20 | 75.56 | 83.11 | 91.57 | 68.51 | 92.10 | 71.57 | 96.45 | 48.84 | 56.02 | 87.50 | 36.63 | 91.86 |
| Textures | 84.54 | 54.97 | 82.2 | 59.28 | 74.49 | 84.96 | 78.03 | 72.27 | 72.98 | 78.48 | 89.10 | 53.26 | 53.97 | 87.56 | 35.53 | 92.66 |
| Average | 95.44 | 51.61 | 95.28 | 46.75 | 74.70 | 83.08 | 74.83 | 75.76 | 79.80 | 76.28 | 81.32 | 62.72 | 63.04 | 82.90 | 49.08 | 87.88 |
| **FGSM** | FPR$_{95}$ | AUC | FPR$_{95}$ | AUC | FPR$_{95}$ | AUC | FPR$_{95}$ | AUC | FPR$_{95}$ | AUC | FPR$_{95}$ | AUC | FPR$_{95}$ | AUC | FPR$_{95}$ | AUC |
| TinyImgNet | 94.84 | 48.04 | 89.52 | 54.45 | 81.94 | 75.58 | 51.15 | 85.38 | 83.80 | 69.87 | 71.00 | 77.21 | 84.30 | 64.94 | 74.91 | 80.80 |
| Place365 | 94.10 | 49.78 | 88.51 | 55.26 | 83.65 | 74.79 | 44.84 | 88.46 | 85.87 | 69.53 | 59.99 | 85.25 | 65.15 | 82.79 | 76.45 | 76.95 |
| LSUN | 99.84 | 54.00 | 98.62 | 49.36 | 60.37 | 85.52 | 15.98 | 96.07 | 50.44 | 89.09 | 12.13 | 97.15 | 88.46 | 66.99 | 41.45 | 89.86 |
| LSUN-R | 99.58 | 47.45 | 99.11 | 47.05 | 83.61 | 73.06 | 66.59 | 82.89 | 93.18 | 56.7 | 82.08 | 68.89 | 57.56 | 80.55 | 40.05 | 91.39 |
| iSUN | 99.05 | 49.45 | 98.38 | 48.31 | 81.65 | 74.62 | 74.14 | 78.50 | 93.56 | 56.54 | 86.36 | 64.21 | 60.69 | 80.65 | 42.31 | 90.40 |
| Textures | 85.87 | 54.44 | 79.72 | 60.80 | 78.23 | 81.00 | 66.88 | 74.98 | 65.59 | 78.20 | 78.35 | 63.15 | 66.13 | 82.32 | 39.88 | 91.26 |
| Average | 95.55 | 50.53 | 92.31 | 52.54 | 78.24 | 77.43 | 53.26 | 84.38 | 78.74 | 69.99 | 64.99 | 75.98 | 70.38 | 76.37 | 52.51 | 86.78 |
| **CW** | FPR$_{95}$ | AUC | FPR$_{95}$ | AUC | FPR$_{95}$ | AUC | FPR$_{95}$ | AUC | FPR$_{95}$ | AUC | FPR$_{95}$ | AUC | FPR$_{95}$ | AUC | FPR$_{95}$ | AUC |
| TinyImgNet | 94.13 | 48.67 | 94.21 | 48.22 | 81.94 | 80.26 | 77.98 | 76.06 | 85.98 | 67.09 | 86.58 | 69.65 | 81.98 | 67.91 | 76.72 | 79.81 |
| Place365 | 95.33 | 48.75 | 95.40 | 47.09 | 84.37 | 78.41 | 78.09 | 75.83 | 74.62 | 79.00 | 85.62 | 69.20 | 67.11 | 79.34 | 78.57 | 77.17 |
| LSUN | 99.72 | 55.28 | 99.91 | 42.87 | 58.97 | 88.51 | 39.09 | 90.20 | 11.95 | 96.90 | 33.10 | 92.97 | 85.07 | 67.87 | 38.35 | 91.06 |
| LSUN-R | 99.55 | 49.81 | 99.76 | 40.06 | 78.78 | 83.03 | 90.25 | 72.10 | 69.39 | 83.66 | 88.48 | 67.62 | 57.26 | 82.32 | 35.89 | 92.47 |
| iSUN | 98.86 | 52.10 | 99.60 | 41.99 | 76.90 | 83.07 | 92.34 | 68.82 | 70.17 | 81.92 | 91.47 | 64.55 | 59.84 | 82.38 | 38.90 | 91.09 |
| Textures | 84.68 | 54.87 | 81.70 | 59.35 | 75.64 | 85.09 | 79.24 | 72.43 | 48.21 | 88.19 | 74.33 | 74.53 | 70.55 | 76.55 | 36.45 | 92.33 |
| Average | 95.38 | 51.58 | 95.10 | 46.6 | 76.10 | 83.06 | 76.17 | 75.91 | 60.05 | 82.79 | 76.60 | 73.09 | 70.30 | 76.06 | 50.81 | 87.32 |

Table 9: Full ablation results for different OOD detection methods using *CIFAR-10* as ID samples and the other six datasets as OOD samples. AT denotes adversarial training.

| Model | CIDER | | | | | | MGP | | | | SaGD | | | | | | | |
|---|---|---|---|---|---|---|---|---|---|---|---|---|---|---|---|---|---|---|
| RSAM | ✓ | | ✗ | | ✓ | | ✓ | | ✗ | | ✓ | | ✓ | | ✓ | | ✓ | |
| AT | ✗ | | Jitter | | Jitter | | ✗ | | Jitter | | FGSM | | PGD | | FAB | | CW | |
| **Clean** | FPR95 | AUC | FPR95 | AUC | FPR95 | AUC | FPR95 | AUC | FPR95 | AUC | FPR95 | AUC | FPR95 | AUC | FPR95 | AUC | FPR95 | AUC |
| TinyImgNet | 79.68 | 80.09 | 46.91 | 90.86 | 60.53 | 87.44 | 35.92 | 92.81 | 42.11 | 91.53 | 94.09 | 45.47 | 94.47 | 46.46 | 95.46 | 47.32 | 57.11 | 88.36 |
| places365 | 75.02 | 84.15 | 40.03 | 92.53 | 47.45 | 91.1 | 28.75 | 94.72 | 38.08 | 93.22 | 87.49 | 65.09 | 85.79 | 70.63 | 92.83 | 59.73 | 47.68 | 90.68 |
| LSUN | 4.63 | 98.72 | 8.01 | 98.71 | 7.75 | 98.35 | 8.79 | 98.66 | 35.36 | 95.23 | 92.92 | 67.23 | 99.94 | 20.53 | 78.38 | 70.43 | 9.81 | 98.25 |
| LSUN-R | 73.04 | 87.92 | 43.88 | 93.49 | 62.53 | 91.42 | 23.55 | 96.18 | 23.23 | 96.23 | 97.17 | 43.44 | 78.49 | 84.06 | 83.6 | 69.18 | 60.65 | 89.88 |
| iSUN | 76.72 | 86.91 | 46.22 | 93.26 | 60.23 | 91.46 | 22.07 | 96.38 | 24.36 | 96 | 95.65 | 45.15 | 78.73 | 81.49 | 84.73 | 67.32 | 63.01 | 89.04 |
| Textures | 65.78 | 82.79 | 26.35 | 95.88 | 29.47 | 93.44 | 17.98 | 97.31 | 18.76 | 97.04 | 88.67 | 47.02 | 82.41 | 58.72 | 86.97 | 53.87 | 34.4 | 93.83 |
| AVG | 62.48 | 86.76 | 35.23 | 94.12 | 44.66 | 92.20 | 22.84 | 96.01 | 30.32 | 94.88 | 92.67 | 52.23 | 86.64 | 60.32 | 87.00 | 61.31 | 45.44 | 91.67 |
| **PGD** | FPR95 | AUC | FPR95 | AUC | FPR95 | AUC | FPR95 | AUC | FPR95 | AUC | FPR95 | AUC | FPR95 | AUC | FPR95 | AUC | FPR95 | AUC |
| TinyImgNet | 91.03 | 63.74 | 68.69 | 82.26 | 61.79 | 86.42 | 84.29 | 69.79 | 45.11 | 90.66 | 78.28 | 87.03 | 98.18 | 77.6 | 39.72 | 91.24 | 80.76 | 79.45 |
| places365 | 89 | 66.27 | 67.07 | 82.67 | 50.05 | 90.18 | 79.87 | 73.88 | 36.58 | 93.57 | 77.42 | 83.32 | 95.75 | 76.56 | 47.41 | 90.21 | 80.94 | 80.39 |
| LSUN | 23.3 | 94.76 | 28.77 | 95.36 | 8.01 | 98.36 | 50.34 | 88.29 | 27.23 | 96.29 | 90.39 | 79.9 | 99.99 | 77.5 | 35.36 | 94.17 | 32.73 | 92.8 |
| LSUN-R | 89.7 | 70.55 | 73.11 | 83.92 | 68.49 | 89.12 | 82.48 | 78.3 | 28.2 | 95.41 | 78.54 | 88.93 | 92.99 | 71.54 | 50.19 | 90.57 | 84 | 82.89 |
| iSUN | 91.2 | 68.44 | 74.53 | 83.78 | 65.9 | 89.37 | 82.99 | 78.94 | 27.97 | 95.45 | 77.85 | 88.89 | 93.08 | 71.44 | 53.62 | 88.89 | 86.53 | 81.2 |
| Textures | 80.32 | 66.14 | 42.93 | 92.01 | 31.26 | 92.92 | 59.34 | 82.69 | 20.14 | 96.82 | 80.59 | 84.69 | 93.76 | 77.88 | 30.02 | 91.72 | 56.51 | 86.53 |
| AVG | 77.43 | 71.65 | 59.18 | 86.67 | 47.58 | 91.06 | 73.22 | 78.65 | 30.87 | 94.70 | 80.51 | 85.46 | 95.63 | 75.42 | 42.72 | 91.13 | 70.24 | 83.88 |
| **Jitter** | FPR95 | AUC | FPR95 | AUC | FPR95 | AUC | FPR95 | AUC | FPR95 | AUC | FPR95 | AUC | FPR95 | AUC | FPR95 | AUC | FPR95 | AUC |
| TinyImgNet | 90.28 | 67.06 | 69.52 | 82.41 | 62.72 | 86.75 | 84.29 | 69.79 | 44.19 | 91.08 | 94.29 | 51.95 | 96.79 | 62.86 | 94.97 | 51.46 | 69.48 | 83.41 |
| places365 | 88.16 | 70.16 | 65.37 | 83.87 | 50.17 | 90.58 | 79.87 | 73.88 | 37.94 | 93.45 | 92.68 | 54.13 | 92.78 | 63.18 | 93.44 | 53.77 | 64.39 | 84.65 |
| LSUN | 13.87 | 96.17 | 18.71 | 96.47 | 7.4 | 98.4 | 50.34 | 88.29 | 30.75 | 95.94 | 97.84 | 53.57 | 99.53 | 56.03 | 96.08 | 50.92 | 13.89 | 97.24 |
| LSUN-R | 81.27 | 80.2 | 78.91 | 83.58 | 67.47 | 90.21 | 82.48 | 78.3 | 29.25 | 95.35 | 95.81 | 44.91 | 93.44 | 62.19 | 92.96 | 50.33 | 61.08 | 88.65 |
| iSUN | 83.74 | 79.21 | 81.41 | 82.67 | 64.58 | 90.12 | 82.99 | 78.94 | 29.18 | 95.29 | 96.08 | 45.9 | 94 | 61.3 | 92.94 | 51.5 | 64.94 | 86.97 |
| Textures | 82.66 | 69.03 | 49.49 | 90 | 31.37 | 93.07 | 59.34 | 82.69 | 20.51 | 96.9 | 93.72 | 51.58 | 90.71 | 60.13 | 95.02 | 52.27 | 47.8 | 89.58 |
| AVG | 73.33 | 76.97 | 60.57 | 86.5 | 47.29 | 91.52 | 73.22 | 78.65 | 31.97 | 94.67 | 95.07 | 50.34 | 94.54 | 60.95 | 94.24 | 51.71 | 53.6 | 88.42 |
| **FAB** | FPR95 | AUC | FPR95 | AUC | FPR95 | AUC | FPR95 | AUC | FPR95 | AUC | FPR95 | AUC | FPR95 | AUC | FPR95 | AUC | FPR95 | AUC |
| TinyImgNet | 84.22 | 68.18 | 68.58 | 82.15 | 70.24 | 84.02 | 84.5 | 69.69 | 47.02 | 90.87 | 98.82 | 50.86 | 89.18 | 77.47 | 39.4 | 91.35 | 80.15 | 79.41 |
| places365 | 82.18 | 71.03 | 67.75 | 82.36 | 60.96 | 88.07 | 79.34 | 73.99 | 37 | 91.86 | 98 | 51.91 | 88.14 | 77.67 | 47.47 | 90.35 | 80 | 80.09 |
| LSUN | 14.59 | 96.11 | 29.63 | 95.22 | 8.25 | 98.06 | 49.78 | 88.61 | 27.18 | 96.27 | 97.84 | 51.63 | 87.67 | 77.91 | 35.9 | 93.95 | 32.29 | 92.83 |
| LSUN-R | 78.96 | 76.9 | 73.49 | 83.87 | 80.64 | 87.47 | 81.56 | 78.86 | 28.77 | 95.33 | 99.32 | 49.98 | 87.64 | 77.55 | 50.69 | 90.58 | 83.68 | 82.85 |
| iSUN | 82.06 | 75.38 | 75.45 | 83.57 | 78.07 | 87.62 | 82.58 | 79.32 | 28.62 | 95.35 | 99.05 | 50.45 | 88.11 | 77.79 | 53.61 | 89 | 86.14 | 81.04 |
| Textures | 75.74 | 66.01 | 43.17 | 91.95 | 37.18 | 91.39 | 59.59 | 82.88 | 20.35 | 96.81 | 97.71 | 51.75 | 88.79 | 77.66 | 30.43 | 91.69 | 55.85 | 86.39 |
| AVG | 69.63 | 75.60 | 59.68 | 86.52 | 55.89 | 89.44 | 72.89 | 78.89 | 31.49 | 94.66 | 98.46 | 51.10 | 88.26 | 77.74 | 42.92 | 91.15 | 69.69 | 83.77 |
| **FGSM** | FPR95 | AUC | FPR95 | AUC | FPR95 | AUC | FPR95 | AUC | FPR95 | AUC | FPR95 | AUC | FPR95 | AUC | FPR95 | AUC | FPR95 | AUC |
| TinyImgNet | 92.48 | 68.44 | 84.54 | 73.32 | 61.87 | 87.32 | 90.27 | 60.05 | 54.57 | 88.64 | 56.2 | 92.1 | 18.8 | 95.03 | 99.83 | 60.61 | 79.47 | 80.41 |
| places365 | 88.42 | 75.3 | 79.47 | 77.05 | 49.86 | 90.88 | 84.58 | 67.31 | 44.9 | 91.86 | 45.39 | 91.68 | 27.59 | 95.09 | 99.47 | 75.25 | 82.6 | |
| LSUN | 11.62 | 97.21 | 27.12 | 94.92 | 8.37 | 98.26 | 25.72 | 92.98 | 18.4 | 97.12 | 64.87 | 91.42 | 8.91 | 95.56 | 99.52 | 63.4 | 19.89 | 96.21 |
| LSUN-R | 87.47 | 75.46 | 90.7 | 77.48 | 65.73 | 91.05 | 95.1 | 62.07 | 43.74 | 92.83 | 63.5 | 90.87 | 19.34 | 96.09 | 98.95 | 63.75 | 87.3 | 81.56 |
| iSUN | 88.11 | 75.52 | 88.17 | 78.62 | 63.05 | 91.13 | 93.43 | 65.34 | 42.57 | 93.21 | 61.71 | 90.9 | 17.93 | 96 | 99.16 | 61.27 | 86.81 | 81.14 |
| Textures | 85.6 | 72.94 | 53.3 | 88.17 | 30.23 | 93.37 | 64.2 | 78.66 | 24.27 | 95.98 | 63.53 | 90.23 | 28.92 | 93.39 | 99.49 | 59.3 | 51.52 | 90.18 |
| AVG | 75.62 | 77.48 | 70.55 | 81.59 | 46.52 | 92.00 | 75.55 | 71.07 | 38.08 | 93.27 | 59.20 | 91.20 | 20.25 | 95.19 | 99.46 | 61.23 | 66.71 | 85.35 |
| **CW** | FPR95 | AUC | FPR95 | AUC | FPR95 | AUC | FPR95 | AUC | FPR95 | AUC | FPR95 | AUC | FPR95 | AUC | FPR95 | AUC | FPR95 | AUC |
| TinyImgNet | 81.15 | 78.69 | 68.39 | 82.3 | 61.16 | 87.37 | 83.21 | 70.05 | 45.6 | 90.88 | 94.09 | 45.47 | 94.47 | 46.46 | 40.54 | 91.18 | 60.41 | 87.44 |
| places365 | 75.89 | 83.11 | 67.65 | 82.55 | 48.1 | 91.06 | 79.06 | 73.87 | 38.14 | 93.35 | 86.95 | 65.22 | 85.54 | 70.9 | 48.81 | 90.06 | 51.23 | 89.52 |
| LSUN | 5.3 | 98.58 | 29.76 | 95.2 | 7.97 | 98.34 | 48.85 | 88.78 | 27.33 | 96.28 | 93.75 | 67.12 | 99.94 | 20.53 | 36.44 | 93.95 | 11.02 | 98.06 |
| LSUN-R | 75.89 | 86.71 | 74.52 | 83.8 | 63.9 | 91.33 | 80.23 | 78.93 | 28.66 | 95.37 | 97.17 | 43.44 | 78.49 | 84.06 | 50.71 | 90.48 | 65.36 | 88.82 |
| iSUN | 79.08 | 85.64 | 75.81 | 83.68 | 61.47 | 91.37 | 81.6 | 79.13 | 28.66 | 95.38 | 95.65 | 45.15 | 78.73 | 81.49 | 54.96 | 88.6 | 67.96 | 87.92 |
| Textures | 66.76 | 81.54 | 44.08 | 92.05 | 29.75 | 93.38 | 58.4 | 82.91 | 19.95 | 96.79 | 88.67 | 47.02 | 82.41 | 58.72 | 31.28 | 91.51 | 36.67 | 93.19 |
| AVG | 69.62 | 75.6 | 60.03 | 86.6 | 45.39 | 92.14 | 71.89 | 78.94 | 31.39 | 94.68 | 98.46 | 51.1 | 88.25 | 77.74 | 42.92 | 91.15 | 48.77 | 90.83 |

