# OpenReview forum: "Sharpness-Aware Geometric Defense for Robust Out-Of-Distribution Detection"
_ICLR.cc/2025/Conference — ICLR 2025 Conference Withdrawn Submission_

### Official Review · Reviewer_EcSE · 2024-10-28

**Soundness:** 2
**Presentation:** 3
**Contribution:** 2
**Rating:** 6
**Confidence:** 4

**Summary:**

This paper presents a robust method for out-of-distribution (OOD) detection that effectively separates adversarial in-distribution (ID) samples from OOD ones. It introduces the Sharpness-aware Geometric Defense (SaGD) framework, which smooths the irregular adversarial loss landscape within the projected latent space. By improving the convergence of geometric embeddings, the framework enhances the characterization of ID data, strengthening OOD detection in the presence of adversarial attacks. Additionally, the use of jitter-based perturbations in adversarial training expands the defense against unseen threats. Experimental results demonstrate that the SaGD framework achieves significant improvements in false positive rate (FPR) and area under the curve (AUC) compared to state-of-the-art methods, particularly in distinguishing CIFAR-100 from six other OOD datasets under various attack scenarios.

**Strengths:**

1. It introduces a novel sharpness-aware method for improving OOD detection in adversarial training. The proposed method investigates the combination of Riemannian geometries under adversarial conditions. This expansion of geometry space sharpens the proposed defense against adversarial attacks and avoids reliance on large OOD datasets for auxiliary training.

2. The proposed SaGD sets a new SoTA for OOD detection, excelling in $FPR_{95}$ and AUC metrics, both with or without attacks.

3. It performs ablation experiments to analyze the relations between the minimization of a sharp loss landscape and OOD detection performance under various adversarial conditions.

**Weaknesses:**

1. It should provide a detailed analysis of the computational complexity involved in computing the OOD score. Additionally, it is important to examine how the number of in-distribution (ID) training samples affects the performance of the OOD score, as this can influence the scalability and generalizability of the approach.

2. Choosing an appropriate threshold $\lambda$ for the OOD score can be challenging in real-world applications. The paper should include a clear, practical procedure for determining this threshold to ensure consistent performance across diverse datasets and scenarios.

3. To thoroughly validate the robustness of the proposed defense, it should incorporate adaptive attacks specifically designed to exploit the OOD scoring mechanism. Following the recommendations in [1], it should evaluate the effectiveness of the defense against these adaptive attacks to demonstrate its resilience under targeted adversarial conditions.

[1] Tramer, Florian, et al. "On adaptive attacks to adversarial example defenses." Advances in neural information processing systems 33 (2020): 1633-1645.

**Questions:**

1. Could the authors analyze the computational complexity of computing the OOD score, and how does it scale with the size of the dataset? Additionally, can the authors provide insights into how the number of in-distribution (ID) training samples affects the performance of the method?

2. In practical applications, selecting the threshold $\lambda$ for the OOD score can be challenging. Could the authors elaborate on the procedure for choosing an optimal threshold, especially under varying dataset conditions and deployment scenarios?

3. Could the authors design adaptive attacks that directly target the proposed OOD scoring mechanism and evaluate the proposed defense against such adaptive attacks?

---

### Official Review · Reviewer_DDJG · 2024-11-04

**Soundness:** 2
**Presentation:** 1
**Contribution:** 2
**Rating:** 3
**Confidence:** 4

**Summary:**

The authors propose a sharpness-aware method for improving OOD detection in adversarial training. Specifically, a multi-geometry projection network is trained to extract the hypersphere and hyperbolic features using jitter-based adversarial samples. Moreover, the network is optimized by sharpness-aware loss minimization using RSAM. Extensive experiments demonstrate the effectiveness of the proposed method. However, I have some concerns about this paper. My detailed comments are as follows.

**Strengths:**

1.	The authors investigate various adversarial attacks on different OOD detection approaches. Extensive experiments demonstrate the effectiveness of the proposed method.
2.	They introduce Jitter-based perturbation in adversarial training to extend the defense ability against unseen attacks.
3.	They employ Multi-Geometry Projection (MGP) and Riemannian Sharpness-aware Minimization (RSAM) for the OOD detection.

**Weaknesses:**

1.	My first concern is the reasonability of the research setting. The paper presents a method to classify adversarial examples as in-distribution (ID) samples in the context of out-of-distribution (OOD) detection. However, I find the rationale for this setting questionable for two main reasons:
* Adversarial examples, by design, deviate significantly from the natural data distribution, even if they remain close in image space. Treating them as OOD samples aligns with standard OOD detection objectives, as these samples no longer represent the semantic consistency of ID data.
* Detecting adversarial examples as OOD is practically advantageous, as it helps prevent their influence on model predictions. For most applications, identifying adversarial samples as OOD is a more effective way to mitigate potential risks, while treating them as ID can increase vulnerability to attacks.

2.	The novelty of the methodology is limited. The proposed method appears to be a fusion of the MMEL approach [1] and the RSAM technique [4], denoted as MPG and RSAM, respectively, within the present paper.
3.	The motivations of the introduction for the three components in the approach are not clear. Why do you use MGP, RSAM and Jitter-based perturbation?
4.	The content of Figure 2 appears to have been adapted from Figure 1 in the referenced paper [1].
5.	The significance of the sharp loss landscape seems to be self-evident, as it has been extensively explored in the existing literature [2, 3]. Regrettably, I fail to discern any novel contribution from the current paper in this regard.

[1] Learning Multi-Manifold Embedding for Out-Of-Distribution Detection

[2] Sharpness-Aware Minimization for Efficiently Improving Generalization

[3] Detecting Adversarial Samples through Sharpness of Loss Landscape

[4] Riemannian SAM: Sharpness-Aware Minimization on Riemannian Manifolds

**Questions:**

1.	Is the hyperspherical geometry learning a type of adverbial defense method? I did not see the related description in the section “ADVERSARIAL DEFENSES”.
2.	Why not use other attack methods to perform adversarial training?
3.	How to obtain the class prototype $\mu_k$ ?

---

### Official Review · Reviewer_oonZ · 2024-11-04

**Soundness:** 2
**Presentation:** 2
**Contribution:** 1
**Rating:** 3
**Confidence:** 5

**Summary:**

This paper argues that adversarial examples should be classified as in-distribution samples rather than outliers. The authors imagine out-of-distribution (OOD) detection scenarios where input data may be subject to adversarial attacks. They demonstrate that their proposed method maintains robust OOD detection performance even when the data contains adversarial perturbations. The authors achieve strong experimental results by incorporating several established techniques in their approach.

**Strengths:**

- The paper presents a novel angle by examining OOD detection under potential adversarial attacks - a scenario that has received limited attention.
- The experimental evaluation is comprehensive and thorough.

**Weaknesses:**

## Should adversarial examples be classified as in-distribution samples rather than outliers?
It is clear that by adding adversarial perturbation, the distribution shifted, why it still should be in-distribution?

## About the scenario
What are some real-world applications where out-of-distribution detection must handle potentially adversarially attacked images?

## About the Contribution
The contribution of this work should be carefully justified. Most of the subsection in section 3 are existing methods. In the introduction section, the authors claim that the smoother regularizer introduced in 3.3 is the key contribution, but I do not agree that this intuition based smoothing regularizer is enough to let this paper be accept.

Given the limited practical relevance of the scenario and modest contributions, I recommend rejection.

**Questions:**

See weaknesses

**Details Of Ethics Concerns:**

N / A

---

> ### Comment · Reviewer_oonZ · 2024-11-26
>
> The discussion deadline is near, I will keep the score if there is no response to this review.

---

### Note · Authors · 2024-11-27

I have read and agree with the venue's withdrawal policy on behalf of myself and my co-authors.